# A study of the associations between social isolation and loneliness with sex-specific cancer risk in the UK Biobank

Jiahao Cheng [1,2,3,5], Runchen Wang[1,2,5], Yi Feng [1,2,5], Shijie Ye[1,4], Hengrui Liang [1,2], Bo Cheng[1,2], Qi Cai[1,2], Shan Xiong [1,2], Yulin Zhao[1,2,3], Xuanzhuang Lu[1,2,3], Qi Zhang[3], Xufeng Zhao[1,4], Juan He[1,3], Peiyu Ma[1,4], Jianxing He [1,2] ✉ & Wenhua Liang [1,2] ✉

## Abstract

**Background** Social isolation, an objective lack of social connections, and loneliness, the subjective distress from perceived social deficits, are established risk factors for poor cancer prognosis. However, their associations with cancer incidence remain unclear. We investigated these associations using UK Biobank data.

**Methods** We analyzed data from 354,537 UK Biobank participants aged 38–73. Participants linked to national health registries, without cancer within one year post-baseline, and with complete exposure and covariate data were included. The primary outcome was cancer incidence. Covariates were classified into demographic, physiological, socioeconomic, lifestyle, and health-related indicators. Cox proportional hazards models were used, with subgroup interaction analysis and mediation analyses performed.

**Results** Here we show that 20,767(5.8%) of participants are isolated and 15,942(4.5%) of participants are lonely. During a median 11.60 years (IQR8.40–12.72) of follow-up, 38,103 participants are diagnosed with cancer. After adjusting for covariates, social isolation is associated with an 8% higher cancer risk(CSHR1.087 95% CI 1.043-1.133; sHR1.073 95% CI 1.029-1.120), while loneliness is not. Social isolation shows a strong interaction by sex (P-interaction<0.01), with isolated females at higher risk than males. Social isolation increases the risk of breast, lung, uterine, ovarian, bladder, and stomach cancers in females, and bladder cancer in males. Socioeconomic factors, health behaviours, and inflammation status largely explain these associations.

**Conclusions** Social isolation is a risk factor for cancer with significant sex and organ-specific effects. Addressing socioeconomic challenges, unhealthy lifestyles, and poor mental well-being through health policies could help reduce cancer risk in isolated populations.

## Plain Language Summary

We investigated whether feelings of loneliness or having few social ties could impact whether a person develops cancer. We studied data from over 350,000 UK adults to see if social isolation (having little social contact) or loneliness (feeling alone) affected cancer risk. Our main finding was that people who were socially isolated had a higher risk of developing cancer, especially women. However, just feeling lonely did not show a direct link to cancer risk. We propose that factors such as income, lifestyle habits, and whether the body has increased levels of inflammation, which is the body's response to infection, might help explain this connection. This suggests that strengthening social connections could be a promising strategy to help prevent cancer. Public health efforts that reduce social isolation, particularly for women, may be an important strategy alongside promoting healthy habits.

Mounting epidemiological evidence underscores the significant role of social determinants of health (SDOH), such as social isolation (SI) and loneliness, in both physical and mental well-being[1], with growing evidence highlighting their overlap with cancer incidence and prevention efforts[2]. While these constructs are frequently conflated, they represent distinct psychosocial phenomena: SI is defined as a lack of social network or community connections, such as having infrequent social contact with family, friends or groups, or living alone, whereas loneliness refers to the subjective discrepancy between an individual's desired and actual social relationships[3,4]. SI and loneliness are widespread, with ~20–25% of

[1]Department of Thoracic Surgery and Oncology, the First Affiliated Hospital of Guangzhou Medical University, State Key Laboratory of Respiratory Disease & National Clinical Research Center for Respiratory Disease, Guangzhou, China. [2]Guangzhou Institute of Respiratory Health, Guangzhou, China. [3]Department of Clinical Medicine, Nanshan School, Guangzhou Medical University, Guangzhou, China. [4]Department of Clinical Medicine, The First Clinical School of Guangzhou Medical University, Guangzhou, China. [5]These authors contributed equally: Jiahao Cheng, Runchen Wang, Yi Feng. ✉e-mail: drjianxing.he@gmail.com; liangwh1987@163.com

**Fig. 1 | Participant flow diagram for the analysis of SI and loneliness in the UK Biobank cohort (baseline 2006–2010).** This diagram illustrates the flow of participants through the study, including enrollment, inclusion/exclusion criteria, and group allocation based on exposure to SI and loneliness. Numbers represent the count of participants at each stage. SI Social Isolation.

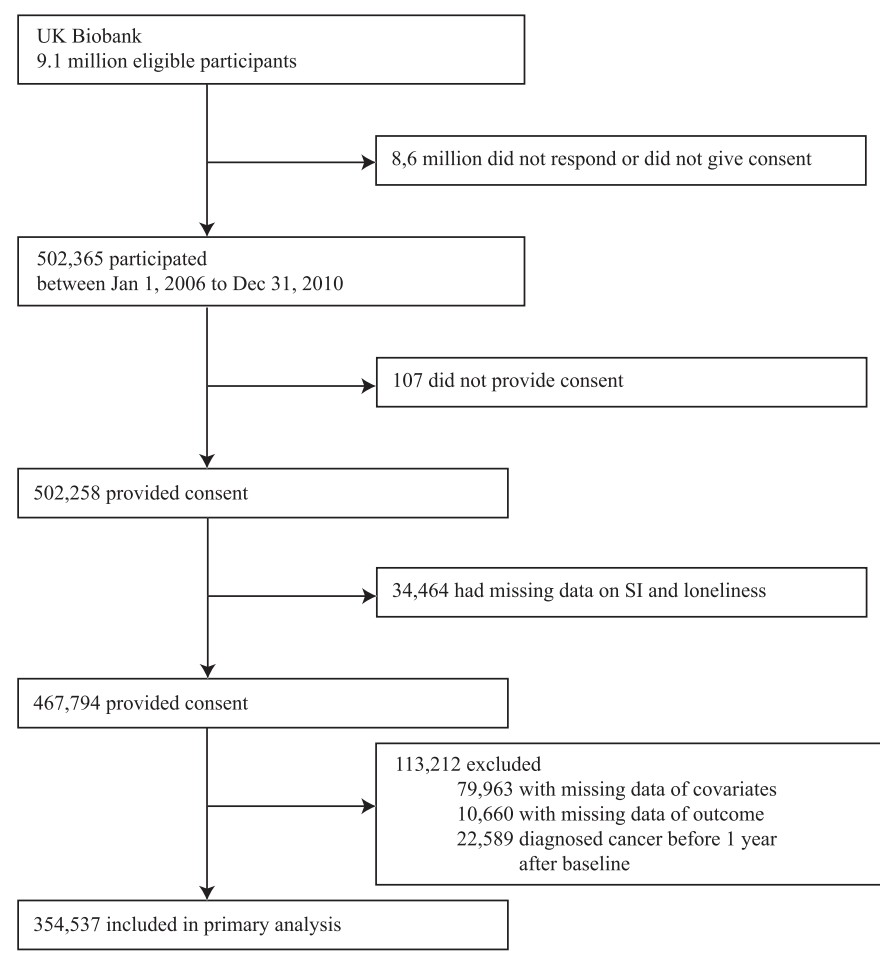

community-dwelling older adults experiencing SI[5], and ~5–15% of adolescents experiencing loneliness[6–8], varying by population and measurement approaches. As issues affecting people of all ages, particularly older adults and adolescents, SI and loneliness are increasingly recognized as priority public health problems[9].

Accumulating evidence positions SI and loneliness not only as risk factors for various diseases, including depression[10], cardiovascular diseases[11], infections[12], and diabetes[13], but also as contributors to higher cancer mortality risk[14–17]. Plausible mechanisms suggest that SI and loneliness may increase cancer risk primarily through physiological alterations and behavioral changes[4,18,19]. Physiologically, SI and loneliness act as chronic stressors[20], activating the hypothalamic-pituitary-adrenal (HPA) axis and leading to sympathetic nervous system (SNS) dysregulation[21]. This triggers hormonal imbalances, such as elevated cortisol levels, and chronic inflammation[22–24], impairing DNA repair, promoting oncogene activation, and facilitating tumor initiation and progression[22,25,26]. Behaviorally, affected individuals often adopt unhealthy habits, such as increased smoking, excessive alcohol consumption, reduced physical activity, poor sleep, and suboptimal diet, all of which are established risk factors for cancers like lung, colorectal, and breast[4,27–29].

However, SI and loneliness may also confer potential protective effects in certain contexts. Reduced social interactions could limit exposure to environmental carcinogens, such as second-hand smoke, alcohol, or infectious agents like HPV, HBV, and H. pylori[12]. Moreover, loneliness's subjective distress might spur adaptive re-engagement, mitigating long-term isolation[30,31]. Despite these complexities, there is still a lack of comprehensive large-scale cohort studies that can systematically and thoroughly demonstrate the relationship between loneliness, SI, and cancer incidence.

This study therefore utilizes prospective data from 502,258 participants in the UK Biobank to examine the independent associations of SI and loneliness with overall and site-specific cancer incidence. Our analyses account for a comprehensive set of potential confounders and explore effect modifications by sex and other demographic factors. Here we show that social isolation is associated with an increased overall cancer risk, particularly in females, while loneliness shows no independent association. We further demonstrate that this association is largely mediated by socio-economic factors, health behaviors, and inflammation status. The findings clarify the distinct roles of these psychosocial factors in cancer development and provide important insights for integrating social health into public health strategies for cancer prevention.

## Method
### Study population
We assessed the association of loneliness and SI on cancer using the UK Biobank data, a prospective cohort study that recruited 502,258 participants (5.4% of the eligible population)[32] in the UK between January 1, 2006. and December 31, 2010. (Fig. 1). All participants provided written informed consent for data collection, biological sampling, and record linkage. Baseline data were collected through questionnaires, physical measurements, and biological samples, with long-term follow-up via medical and health records. Participants diagnosed with cancer within 1 year after baseline or with missing data were excluded for complete case analysis.

### Loneliness and SI scale
The SI scale used by the UK Biobank was constructed from three questions: (1) "Including yourself, how many people are living together in your household? Include those who usually live in the house such as students living away from home during term time, partners in the armed forces or professions such as pilots" (1 point for living alone); (2) "How often do you visit friends or family or have them visit you?" (1 point for friends and family

visit less than once a month); and (3) "Which of the following [leisure/social activities] do you engage in once a week or more often? You may select more than one" (1 point for no participation in social activities at least weekly). After scoring 0–3 points, individuals with 2 or 3 were defined as socially isolated, while those with 0 or 1 were not. Similar scales have been used previously in other UK studies[4,12]. Loneliness was assessed with two questions: "Do you often feel lonely?" (0 = no, 1 = yes) and "How often are you able to confide in someone close to you?" (0 = almost daily to once every few months; 1 = never or almost never). An individual was defined as lonely if he or she responded positively to both questions. Similar questions are included in scales such as the revised UCLA Loneliness Scale, the most commonly used multi-item measure of loneliness in population surveys[33]. The measures of SI and loneliness were considered independent exposures, with no overlap or weighting of responses.

### Outcome ascertainment

The primary outcome of the study was the incidence of composite and specific cancer subtypes, identified using ICD-10 codes (C01–C97) from the UK national cancer registry. Total cancer cases included all cancers except non-melanoma skin cancer (C44). Follow-up began at UK Biobank enrollment and continued until a cancer diagnosis, death, or the end of the study period (November 11, 2022), whichever occurred first. The first diagnosed cancer of each participant was considered their primary cancer. Individuals who remained cancer-free during follow-up were designated as the control group, with their data censored at the time of death or the end of the follow-up period. To account for competing risks, non-cancer deaths were treated as competing events (status = 2). Non-cancer death was specified as the competing event for CIFs, mitigating overestimation of risks relative to KM estimates (Supplementary Table 1-1). To ensure analytical robustness, we focused on cancer types with more than 100 cases in the individual cancer analysis (Supplementary Table 3). When available, tumor histology coding was used to stratify cancer diagnoses into common histological subtypes (Supplementary Figs. 1, 2).

### Mediator ascertainment

Based on the evidence of potential pathway, ten inflammation-related biomarkers were selected as potential mediators (Supplementary Tables 4, 5)[26]. In the UK Biobank, blood tests were performed on participants with informed consent at baseline recruitment. Blood samples of about 4 ml were collected, separated by composition, stored in a refrigerator at −80 °C, and analyzed within 24 h using a Beckman Coulter LH750 instrument (https://biobank.ndph.ox.ac.uk/ukb/ukb/docs/haematology.pdf). The blood biomarkers with rigorous quality checks have undergone external validation (https://biobank.ndph.ox.ac.uk/showcase/showcase/docs/serum_biochemistry.pdf). Inflammation-related biomarkers included leukocyte count, neutrophil count, monocyte count, lymphocyte count, platelet count and C reactive protein. In addition, we also included the four systemic inflammation markers: lymphocyte to monocyte ratio, neutrophil to lymphocyte ratio, platelet to lymphocyte ratio and systemic immune inflammation index[34].

### Covariates ascertainment

We considered the following characteristics as potential covariates: age (continuous, years), sex (male/female), race (White/Non-White), mean arterial pressure (continuous, mmHg; calculated as $[SBP + (2 \times DBP)]/3$), BMI (kg/m²), UK assessment center (England/Scotland/Wales), smoking status (never/previous/current), alcohol status (never/previous/current), hand grip strength (kg), day exposure (hours/day), diet score (continuous), depressive mood (high/low/unreported), sleep score (range 0–5), Townsend deprivation index (continuous), education (college/university degree or not), employment status (employed/retired/other), family cancer history (yes/no), overall health rating (scale 1–4), household income (low <£18,000, medium £18,000–£51,999, high ≥£52,000, unreported). The unreported group is comprised of non-responders who selected "do not know" or "prefer not to answer". Detailed definitions, Data-Fields, and sources are

provided in Supplementary Table 6. To address potential multicollinearity among covariates, we computed Pearson correlation coefficients, variance inflation factors (VIF), tolerance values, and condition numbers. Diagnostic criteria included: correlation coefficient r > 0.5, VIF > 2, tolerance <0.5, and condition number >5 (Supplementary Tables 7, 8). The overall condition number for the original covariate correlation matrix was 6.47, indicating low system-wide collinearity.

### Statistics analysis

Individuals reporting SI or loneliness at baseline were stratified by the exposure (Table 1) or sex group (Supplementary Table 9). Crude incidence was assessed using unadjusted incidence rate ratios (IRRs) per 10,000 person-years and 5-year Cumulative Incidence Functions (CIFs) estimated via the Aalen-Johansen method to evaluate the separate and combined effects of SI and loneliness. We further generated the KM curves and CIF curves in overall and sex stratified cohort to visualize unadjusted event probabilities. For main analyses, we employed cause-specific Cox models (CSHR) as primary for etiological estimates, with Fine-Gray competing risks models (sHR) synchronously reported as main findings, using attained age as the time scale and non-cancer death as a competing event. Multi-variable models were generated for the analysis: Model 1 adjusted for age, sex, and race/ethnicity; Model 2 further adjusted for assessment center, sun exposure time, socioeconomic status (Townsend deprivation score), smoking status, and alcohol use. Model 3 further adjusted for BMI (Continuous), grip strength, family history of cancer, MAP, overall health rating, healthy diet score, healthy sleep score and depressive mood. The population attributable fraction (PAF) was calculated using CSHR from Model 3 to estimate the preventable proportion of cancer cases. No violations of the proportional hazards assumption were observed for the exposures using Schoenfeld residuals (Supplementary Fig. 3).

To scrutinize the extent to which baseline biological, behavioral, socioeconomic, psychological and health-related risk factors explained the associations, percentage of excess risk mediated (PERM) was calculated for the following mechanisms: (1) Socioeconomic Status (Townsend Deprivation Index); (2) Health Behaviors (Alcohol Consumption, Sleep Score, Diet Score, Physical Activity and Smoking); (3) Physical Health Indicators (BMI, Physical Health Indicators, MAF); (4) Mental Health (Depress Mood); (5) Overall Health Indicators (Overall Health Rating); (6) Family History (Family Cancer History) and (7) Geographical and Regional Factors (UK Assessment Center). PERM was calculated using the following formula: PERM = [CSHR(age, sex and ethnicity adjusted) −CSHR(age, sex, and ethnicity and risk factor adjusted)]/[CSHR(age, sex and ethnicity adjusted) − 1] × 100[4]. Subgroup analyses were stratified by sex, age (0–49, 50–59, 60+), education, employment, income, alcohol consumption, and smoking status. Multiplicative and additive interaction effects were assessed using CSHR model, enabling a comprehensive evaluation of cancer risks associated with SI and loneliness across different demographic and lifestyle groups[35]. We further analyzed the association between SI and loneliness with specific cancer risks across stratified subgroups. Survival curves for total/sex strata were generated as CIFs to integrally evaluate their effect on cancer incident.

To robustly evaluate potential mediation pathways underlying the associations between SI and cancer incidence, we employed a causal mediation analysis framework using the regression-based approach developed by Valeri and VanderWeele[36]. For each mediator, we fitted a cause-specific Cox proportional hazards model for the outcome (cancer incidence, with attained age as the time scale) and a generalized linear model for the mediators, including inflammatory markers, blood cells, female hormone factors, and menstrual status. All models were adjusted for confounders consistent with the primary multivariable Model 3. Bonferroni correction was applied for multiple mediators in subgroup analyses to control the family-wise error rate. Analyses were conducted for the overall population, stratified by sex, and for site-specific cancers.

We performed several sensitivity analyses, including interpolate missing data of covariates to eliminate the impact of population deletion,

**Table. 1 | Baseline characteristics of the study population (baseline 2006–10)**

| Variable label | Overall (*N* = 354,537) | No SI (*N* = 333,896) | SI (*N* = 20,641) | *P*-value (SI) | No loneliness (*N* = 338,595) | Loneliness (*N* = 15,942) | *P*-value (Lon) |
|---|---|---|---|---|---|---|---|
| Age, years | 56.34 ± 8.06 | 56.28 ± 8.08 | 57.25 ± 7.64 | 1.78e−68 | 56.36 ± 8.06 | 55.85 ± 7.89 | 3.32e−15 |
| Sex, Male, *n* (%) | 163,961 (46%) | 154,730 (46%) | 9231 (45%) | 6.17e−06 | 155,732 (46%) | 8229 (52%) | 5.36e−44 |
| Ethnicity, White, *n* (%) | 340,360 (96%) | 320,660 (96%) | 19,700 (95%) | 2.51e−05 | 325,253 (96%) | 15,107 (95%) | 3.65e−16 |
| Assessment center | | | | 2.04e−04 | | | 1.04e−01 |
| English | 325,924 (92%) | 306,887 (92%) | 19,037 (92%) | | 311,340 (92%) | 14,584 (91%) | |
| Scotland | 12,992 (4%) | 12,189 (4%) | 803 (4%) | | 12,378 (4%) | 614 (4%) | |
| Wales | 15,621 (4%) | 14,820 (4%) | 801 (4%) | | 14,877 (4%) | 744 (5%) | |
| Employment | | | | 2.97e−237 | | | 5.92e−311 |
| Employed | 211,397 (60%) | 200,228 (60%) | 11,169 (54%) | | 202,577 (60%) | 8820 (55%) | |
| Others | 28,422 (8%) | 25,534 (8%) | 2888 (14%) | | 25,880 (8%) | 2542 (16%) | |
| Retired | 114,718 (32%) | 108,134 (32%) | 6584 (32%) | | 110,138 (33%) | 4580 (29%) | |
| College/university degree, *n* (%) | 178,566 (50%) | 169,628 (51%) | 8938 (43%) | 4.49e−97 | 171,749 (51%) | 6817 (43%) | 6.64e−86 |
| Income | | | | 2.64e−2348 | | | 5.58e−477 |
| High | 87,548 (25%) | 85,970 (26%) | 1578 (8%) | | 85,212 (25%) | 2336 (15%) | |
| Low | 64,489 (18%) | 55,433 (17%) | 9056 (44%) | | 59,569 (18%) | 4920 (31%) | |
| Medium | 162,895 (46%) | 154,942 (46%) | 7953 (39%) | | 156,093 (46%) | 6802 (43%) | |
| Unknown | 39,605 (11%) | 37,551 (11%) | 2054 (10%) | | 37,721 (11%) | 1884 (12%) | |
| Smoking status | | | | 3.95e−539 | | | 5.74e−200 |
| Current | 35,339 (10%) | 31,231 (9%) | 4108 (20%) | | 32,660 (10%) | 2679 (17%) | |
| Never | 195,266 (55%) | 185,688 (56%) | 9,578 (46%) | | 187,548 (55%) | 7718 (48%) | |
| Previous | 123,932 (35%) | 116,977 (35%) | 6,955 (34%) | | 118,387 (35%) | 5545 (35%) | |
| Alcohol status | | | | 2.41e−419 | | | 3.49e−73 |
| Current | 329,778 (93%) | 312,023 (93%) | 17,755 (86%) | | 315,480 (93%) | 14,298 (90%) | |
| Never | 13,025 (4%) | 11,812 (4%) | 1213 (6%) | | 12,274 (4%) | 751 (5%) | |
| Previous | 11,734 (3%) | 10,061 (3%) | 1673 (8%) | | 10,841 (3%) | 893 (6%) | |
| Family cancer history, *n* (%) | 111,083 (31%) | 104,462 (31%) | 6621 (32%) | 1.82e−2 | 105,974 (31%) | 5109 (32%) | 4.71e−02 |
| Townsend deprivation score | −1.48 ± 2.97 | −1.59 ± 2.90 | 0.23 ± 3.45 | 4.07e−1072 | −1.52 ± 2.94 | −0.59 ± 3.36 | 2.48e−250 |
| Sun exposure time, hours/day | 2.83 ± 1.89 | 2.83 ± 1.88 | 2.79 ± 2.04 | 1.40e−02 | 2.83 ± 1.88 | 2.86 ± 2.09 | 7.80e−02 |
| Diet score | 2.18 ± 1.21 | 2.18 ± 1.22 | 2.21 ± 1.21 | 9.46e−5 | 2.19 ± 1.21 | 2.01 ± 1.20 | 3.67e−74 |
| Sleep score | 2.39 ± 0.75 | 2.40 ± 0.74 | 2.19 ± 0.82 | 4.11e−258 | 2.40 ± 0.74 | 2.04 ± 0.85 | 4.68e−556 |
| Depress mood | | | | 6.97e−386 | | | 1.16e−2195 |
| High | 15,551 (4%) | 13,458 (4%) | 2093 (10%) | | 12,366 (4%) | 3185 (20%) | |
| Low | 328,107 (93%) | 310,313 (93%) | 17,794 (86%) | | 316,201 (93%) | 11,906 (75%) | |
| Unknown | 10,879 (3%) | 10,125 (3%) | 754 (4%) | | 10,028 (3%) | 851 (5%) | |
| BMI (kg/m²) | 27.26 ± 4.86 | 27.22 ± 4.80 | 27.83 ± 5.73 | 1.05e−50 | 27.21 ± 4.82 | 28.25 ± 5.61 | 1.88e−115 |
| MAP (mmHg) | 101.23 ± 12.47 | 101.20 ± 12.45 | 101.61 ± 12.77 | 9.00e−06 | 101.24 ± 12.47 | 100.91 ± 12.43 | 1.01e−03 |
| Grip strength, kg | 31.00 ± 11.00 | 31.11 ± 11.01 | 29.15 ± 10.79 | 6.58e−140 | 31.02 ± 10.99 | 30.56 ± 11.35 | 4.54e−07 |
| Overall health rating | 2.90 ± 0.73 | 2.92 ± 0.72 | 2.62 ± 0.84 | 2.36e−525 | 2.92 ± 0.72 | 2.47 ± 0.84 | 2.06e−839 |

Baseline variates were presented as means ± standard error or median (interquartile range) for continuous variables and frequency (percentages) for categorical variables. Continuous variables were assessed for statistical differences using two-sample *T* tests, ANOVA tests, or Mann–Whitney U tests. Categorical variables were evaluated for differences using the χ2 test.

*ANOVA* Analysis of Variance, *BMI* Body Mass Index, *MAP* Mean Arterial Pressure, χ² Chi-Squared.

excluding participants who developed cancer within the first two years of follow-up to assess the potential reverse causality, and adding one additional day of follow-up for participants lost to follow-up on the same day to address possible bias caused by early exclusion. Meanwhile, to account for the competing effect of death, we further performed a competing risk model. Multiple testing was corrected using the bonferroni method, with adjusted *P*-values below 0.05 considered significant. Corrections were applied separately for SI and loneliness exposures, focusing on subgroup-specific analyses. Bonferroni adjustments for specific cancer types or the exposure variables themselves were not applied, as these analyses were exploratory in nature and should be interpreted with caution. Statistical analyses were performed using R (version 4.1.2), and statistical significance was defined as $P < 0.05$ for two-sided tests.

## Statistics and reproducibility
This study utilized data from the UK Biobank, a large prospective cohort study. The analysis was conducted on the complete case cohort derived from the initial recruitment of 502,258 participants. Participants diagnosed with cancer within one year after baseline or those with missing baseline data were excluded. In this epidemiological design, each participant is considered an independent biological replicate.

All statistical analyses were performed using R (version 4.1.2)[37]. Statistical significance was defined as $P < 0.05$ for two-sided tests. For analyses involving multiple testing, such as in the mediation and subgroup analyses, the Bonferroni correction method was applied to control the family-wise error rate, with adjusted *P*-values below 0.05 considered significant. Crude incidence rates and 5-year Cumulative Incidence Functions (CIFs) were initially assessed using the Aalen-Johansen method. Primary etiological estimates were obtained using cause-specific Cox proportional hazards models (CSHR), with Fine-Gray competing risks models (sHR) synchronously reported, using non-cancer death as a competing event.

The reproducibility and robustness of the findings were ensured through a comprehensive set of sensitivity analyses and subgroup evaluations. To address potential reverse causality, a sensitivity analysis was performed by excluding participants who developed cancer within the first two years of follow-up. The potential impact of population deletion was assessed using sensitivity analysis that interpolated missing data of covariates. The proportional hazards assumption for the exposures was formally verified using Schoenfeld residuals (Supplementary Fig. 3). Multiple modeling approaches (CSHR and Fine-Gray) and different levels of covariate adjustment (Model 1 to Model 3) were used to confirm the stability of the associations.

## Inclusion & ethics statement
The UK Biobank participants are predominantly of White European ancestry and aged between 40 and 69 at recruitment. While this provides a large and deeply phenotyped cohort for discovery, the relative lack of genetic and ancestral diversity may limit the generalizability of our findings to other populations. We acknowledge this limitation and encourage future validation in more diverse cohorts. Ethical Approval and Data Access Participants in the UK Biobank gave written informed consent for their data to be used for health-related research and for linkage to their health records. The UK Biobank study has approval from the North West Multi-center Research Ethics Committee (MREC) as a Research Tissue Bank (RTB) (references 11/NW/0382, 16/NW/0274, and 21/NW/0157). This RTB approval covers all researchers using the resource for approved projects; therefore, separate ethical approval from the authors' local Institutional Review Board (IRB) was not required for this specific study, as it involved the analysis of de-identified data. Access to the UK Biobank data was granted under Application Number 99157.

## Results
### Characteristics of participants
A total of 354,537 participants (mean[SD] age, 56.34[8.06] years; 46% male) were assessed in a median follow-up time of 11.60 years [IQR 8.40–12.72]

(Fig. 1), most participants were white 340,360 (96%). 20,641 (5.8%) had an exposure of SI, and 15,942 (4.5%) had an exposure of loneliness (Table 1, Supplementary Table 9). Compared with participants without SI, participants with SI exposure were more likely to have lower socioeconomic status, higher BMI, and higher smoking, diet score. They were less likely to have a college education, report a high sleep quality, be in good health, or consume alcohol frequently.

### Separate and joint association of SI and loneliness with cancer risk
In tabular analyses, SI was associated with higher cancer incidence (5 Year CIF 4.304% 95% CI 4.025–4.583%, IRR1.211 95% CI 1.164–1.261, Excess Incidence 21.52 95% CI 21.08–21.96), while loneliness showed no significant effect (5 Year CIF 3.619% 95% CI 3.327–3.911%, IRR1.020 95% CI 0.972–1.071). In the joint effect, as the degree of isolation and loneliness increases, the risk of cancer incidence rises (Table 2, Supplementary Fig. 4). After adjusting for covariates (Model3), SI was significantly associated with higher cancer risk (CSHR 1.087 95% CI 1.043–1.133; sHR 1.073 95% CI 1.029–1.120), whereas loneliness showed no significant association (CSHR 0.977 95% CI 0.930–1.026; sHR 0.974 95% CI 0.925–1.025). A progressively increased risk of cancer was observed with increasing levels of SI and loneliness (*P* trend = 0.002(Cox model), 0.008 (Competing risks model)), and no interaction between SI and loneliness was observed (*P* interaction = 0.397(Cox model), 0.499 (Competing risks model)) (Fig. 2).

### Proportions of the SI-cancer incident association attributable to socioeconomic, behavioral, physical health, mental health, overall health, family history and geographical factors
The association between SI and cancer incidence remained robust after adjusting for multiple covariates, with key contributors including socioeconomic factors (34.5% (Cox model); 35.1% (Competing risks model)), health behaviors (30.9% (Cox model); 33.1% (Competing risks model)), and overall health (15.2% (Cox model); 14.4% (Competing risks model)), with a total PERM of 52.8% (Cox model); 53.5% (Competing risks model) (Fig. 3A). In female participants, SI continued to show a significant association with cancer risk after full adjustment (CSHR 1.161 95% CI 1.096–1.229; sHR 1.156 95% CI 1.091–1.225), with key contributors including health behaviors (24.5% (Cox model); 24.4% (Competing risks model)) socioeconomic factors (22.8% (Cox model); 22.6% (Competing risks model)), and overall health (9.7% (Cox model); 9.1% (Competing risks model)), with a total PERM of 41.7% (Cox model); 41.0% (Competing risks model) (Fig. 3B). In contrast, neither loneliness in female participants nor SI in male participants showed a significant association with cancer risk after adjustment (Supplementary Fig. 5B,C). Additionally, loneliness in overall and male participants was not significant even with minimal adjustment (Supplementary Fig. 1A, D).

### The association between SI, loneliness and cancer risk in strata analysis
Socially isolated participants showed substantially higher cumulative cancer incidence over 14 years across sex stratified cohort (females: CIF *P* < 0.001, KM *P* < 0.001; males: CIF *P* = 0.005, KM *P* < 0.001) (Supplementary Fig. 4C, E). Loneliness showed modest sex-stratified elevations in crude assessment (females: CIF *P* = 0.021, KM *P* = 0.013; males: CIF *P* = 0.028, KM *P* = 0.078) (Supplementary Fig. 4D, F).

Figure 4 illustrates stratified analyses after Model3 adjustment, with distinct columns for SI and loneliness and interaction metrics. After adjustment, SI significantly elevated cancer risk in females (CSHR1.16, 95% CI 1.096–1.229; sHR1.156, 95% CI 1.091–1.225), driven by strong multiplicative and additive interactions with sex. No other significant interactions were emerged for SI. Nonetheless, some specific population remains significant post adjustment: older individuals (aged ≥59) (CSHR1.090 95% CI 1.034–1.149; sHR1.071 95% CI 1.013–1.132), participants with medium income (CSHR1.112 95% CI 1.039–1.191; sHR1.107 95% CI 1.035–1.184),

**Table. 2 | Cancer incidence rate ratios and excess incident based on SI and loneliness**

| Group | Exposure | N incident | Cases | Person years | Incident per 10,000 person years | Five year CIF 95% CI | IRR 95% CI | Excess incidence 95% CI |
|---|---|---|---|---|---|---|---|---|
| Overall | | 354,537 | 38,103 | 3,700,341.18 | 102.972 | 3.529% (3.467%, 3.59%) | | |
| Isolation | Unexposed | 333,896 | 35,521 | 3,490,888.35 | 101.753 | 3.481% (3.418%, 3.543%) | 1 [Reference] | 0 [Reference] |
| | Exposed | 20,641 | 2582 | 209,452.82 | 123.274 | 4.304% (4.025%, 4.583%) | 1.211 (1.164–1.261) | 21.52 (21.079–21.961) |
| Loneliness | Unexposed | 338,595 | 36,372 | 3,535,419.38 | 102.879 | 3.524% (3.462%, 3.587%) | 1 [Reference] | 0 [Reference] |
| | Exposed | 15,942 | 1731 | 164,921.80 | 104.959 | 3.619% (3.327%, 3.911%) | 1.02 (0.972–1.071) | 2.08 (1.586–2.574) |
| Joint effect | Rank1 | 320,660 | 34,137 | 3,353,039.99 | 101.809 | 3.482% (3.418%, 3.546%) | 1 [Reference] | 0 [Reference] |
| | Rank2 | 13,236 | 1384 | 137,848.36 | 100.400 | 3.455% (3.141%, 3.768%) | 0.986 (0.935–1.041) | −1.409 (−1.948 to −0.87) |
| | Rank3 | 17,935 | 2235 | 182,379.39 | 122.547 | 4.287% (3.988%, 4.585%) | 1.204 (1.153–1.256) | 20.738 (20.266–21.209) |
| | Rank4 | 2706 | 347 | 27,073.43 | 128.170 | 4.421% (3.641%, 5.201%) | 1.259 (1.133–1.399) | 26.361 (25.165–27.557) |

Incidence per 10,000 person-years was calculated by dividing the number of cancer cases by the total person-years for each group. The 5-year CIF was used to estimate the probability of developing cancer within 5 years. IRRs were calculated by comparing the cancer incidence rate for each group exposed to SI and/or loneliness to the reference group; Excess incidence represents the additional cancer cases per 10,000 person-years observed in the exposed groups compared to the unexposed groups. Rank 1: no loneliness + no isolation; Rank 2: loneliness + no isolation; Rank 3: no loneliness + isolation; Rank 4: loneliness + isolation.
*SI* Social Isolation, *CI* Confidence Interval, *CIF* Cumulative Incidence Function, *IRR* Incidence Rate Ratio, *N* Number.

those without a college degree (CSHR1.101 95% CI 1.045–1.161; sHR1.068 95% CI 1.031–1.147), retirees (CSHR1.111 95% CI 1.044–1.183; sHR1.091 95% CI 1.024–1.163), employed individuals (CSHR1.083 95% CI 1.019-1.152; sHR1.080 95% CI 1.010–1.154), and current drinkers all exhibited an increased cancer risk (CSHR1.087 95% CI 1.040–1.136; sHR1.079 95% CI 1.033–1.127) (Fig. 4, Supplementary Table 10).

Although loneliness had no significant effect on the overall population after adjustment, significant multiplicative and additive interactions were observed with age, sex, income, education, and employment status. Notably, younger individuals (aged 0–49) showed decreased cancer risk with loneliness (CSHR0.792 95% CI 0.764–0.931; sHR0.791 95% CI 0.682–0.917). As did those employed individuals (CSHR0.888 95% CI 0.823–0.959; sHR0.892 95% CI 0.822–0.968) and unreported income participants (CSHR0.831 95% CI 0.716–0.966; sHR0.835 95% CI 0.714–0.977), all showing stable reduced cancer risk (Fig. 4, Supplementary Table 11).

### SI and loneliness increase specific cancer risk in various populations

After adjusting for age and ethnicity, SI and loneliness were associated with higher cancer risk in both male and female participants (Supplementary Figs. 6, 7). In the fully adjusted model (Model3), SI increased the risk of bladder cancer (CSHR1.50 95% CI 1.12–2.01) in male participants (Supplementary Fig. 8A). In females, SI was linked to higher risks of several cancers, notably stomach cancer (CSHR1.84 95% CI 1.16–2.93) and bladder cancer (CSHR1.68 95% CI 1.08–2.62). Ovarian cancer (CSHR1.52 95% CI 1.16–1.99), uterine (CSHR1.36 95% CI 1.09–1.70), breast (CSHR1.13 95% CI 1.03–1.25), and lung cancers (CSHR1.28 95% CI 1.09–1.51) also showed an increased risk (Supplementary Fig. 8B). No site-specific risks emerged for loneliness exposure after adjustment (Supplementary Fig. 8). Histological subtype analyses revealed no significant SI associations in either sex (Supplementary Figs. 1, 2).

Figure 5 depicts site-specific cancer risks for SI and loneliness in heatmaps, with red shading for increased risk (CSHR > 1), green for decreased (CSHR < 1), and darker shades for stronger significance ($P < 0.05$ marked as a; Bonferroni-adjusted as b). This heatmap showed the relationship between various population characteristics and cancer risk, showing that SI significantly affects cancer risk across several groups. SI was significantly associated with an increased risk of five cancer after adjustment: ovarian, uterine, bladder, lung, and mouth cancers. The CSHRs ranged from 1.52 (95% CI 1.16–1.99) to 7.60 (95% CI 2.21–26.14). Ovarian cancer risk increased notably in female participants (CSHR1.52 95% CI 1.16–1.99), current drinker (CSHR1.57 95% CI 1.18–2.09), and participants with unreported income (CSHR2.78 95% CI 1.45–5.33). Uterine cancer risk increased in employed (CSHR1.75 95% CI 1.3–2.34), medium income (CSHR1.71 95% CI 1.28–2.28), and college/university degree participants (CSHR1.77 95% CI 1.3–2.42). Meanwhile, current drinker (CSHR1.63 95% CI 1.26–2.1), elder (age >59) (CSHR1.74 95% CI 1.32–2.29), employed participants (CSHR1.90 95% CI 1.31–2.75), and college/university degree (CSHR2.09 95% CI 1.44–3.04) have an increased cancer risk of bladder cancer. High income population had a notably higher risk of mouth cancer (CSHR7.60 95% CI 2.21–26.14). Cancers like biliary duct, bladder, breast, cervix, colorectal, esophageal, lung, lymphoma, liver, leukemia, kidney, melanoma, ovarian, pancreatic, prostate, soft tissue, stomach, testis, thyroid, tonsil, and uterine cancer were associated with SI and(or) loneliness in other subgroups, but after correction, these associations were not statistically significant (Supplementary Data 3).

### Mediation analysis

The mediation analysis revealed that inflammatory markers and specific hormonal factors partly mediated the association between SI and cancer incidence. Female sex hormones and menstrual status had no significant impact on overall cancer risk (Supplementary Fig. 9A). However, SHBG significantly mediated the association between SI and uterine cancer in female participants after adjustment, explaining −7.3% of the effect (IE-0.0169 95% CI −0.0248–−0.0091). Similarly, SHBG mediated the effect

| Group Variable | N | Cases per 10000 Person Years | Model1 HR(95%CI) P value | Model2 HR(95%CI) P value | Model3 HR(95%CI) P value | PAF(95%CI) |
|---|---|---|---|---|---|---|
| Overall | 354537 | 103 | | | | |
| Isolation | | | | | | 0.999 [0.514 - 1.465] |
| | | | | | | 1.861 [0.765 - 2.912] |
| Unexpose | 333896 | 102 | 1.000 [Reference] | 1.000 [Reference] | 1.000 [Reference] | |
| | | | 1.000 [Reference] | 1.000 [Reference] | 1.000 [Reference] | |
| Expose | 20641 | 123 | 1.184 [1.138-1.232] 2.22e-16 | 1.091 [1.047-1.137] 3.28e-05 | 1.087 [1.043-1.133] 7.75e-05 | |
| | | | 1.158 [1.110-1.208] 7.49e-12 | 1.076 [1.031-1.123] 7.12e-04 | 1.073 [1.029-1.120] 1.04e-03 | |
| loneliness | | | | | | -0.259 [-0.822 - 0.277] |
| | | | | | | -0.601 [-1.822 - 0.559] |
| Unexpose | 338595 | 103 | 1.000 [Reference] | 1.000 [Reference] | 1.000 [Reference] | |
| | | | 1.000 [Reference] | 1.000 [Reference] | 1.000 [Reference] | |
| Expose | 15942 | 105 | 1.048 [0.999-1.100] 5.70e-02 | 0.985 [0.938-1.034] 5.42e-10 | 0.977 [0.930-1.026] 5.42e-01 | |
| | | | 1.034 [0.986-1.085] 1.68e-01 | 0.978 [0.930-1.029] 3.95e-01 | 0.974 [0.925-1.025] 3.16e-01 | |
| P for interaction | | | 5.22e-01 | 4.77e-01 | 3.97e-01 | |
| | | | 6.31e-01 | 5.77e-01 | 4.99e-01 | |
| Joint Effect | | | | | | 0.379 [0.146 - 0.608] |
| | | | | | | 0.779 [0.201 - 1.346] |
| Rank1 | 320660 | 102 | 1.000 [Reference] | 1.000 [Reference] | 1.000 [Reference] | |
| | | | 1.000 [Reference] | 1.000 [Reference] | 1.000 [Reference] | |
| Rank2 | 13236 | 100 | 1.019 [0.965-1.075] 4.99e-01 | 0.968 [0.917-1.022] 2.46e-01 | 0.960 [0.909-1.014] 1.43e-01 | 0.309 [0.119 - 0.496] |
| | | | 1.009 [0.955-1.067] 7.38e-01 | 0.965 [0.910-1.022] 2.24e-01 | 0.960 [0.905-1.018] 1.73e-01 | 0.503 [0.130 - 0.869] |
| Rank3 | 17935 | 123 | 1.176 [1.126-1.227] 1.30e-13 | 1.087 [1.041-1.136] 1.85e-04 | 1.082 [1.036-1.131] 4.29e-04 | 0.369 [0.142 - 0.591] |
| | | | 1.152 [1.100-1.207] 2.91e-09 | 1.074 [1.024-1.126] 3.00e-03 | 1.070 [1.021-1.122] 5.00e-03 | 0.645 [0.166 - 1.115] |
| Rank4 | 2706 | 128 | 1.248 [1.122-1.387] 4.13e-05 | 1.102 [0.990-1.226] 7.40e-02 | 1.097 [0.985-1.221] 9.20e-02 | 0.379 [0.146 - 0.608] |
| | | | 1.200 [1.086-1.327] 3.63e-04 | 1.075 [0.973-1.187] 1.57e-01 | 1.074 [0.972-1.187] 1.63e-01 | 0.779 [0.201 - 1.346] |
| P trend | | | 7.81e-16 | 5.55e-04 | 2.01e-03 | |
| | | | 5.83e-12 | 6.24e-03 | 8.12e-03 | |

Cox Proportional Hazards Regression (red) — Competing Risk Regression (blue)

**Fig. 2 | Separate and joint association of SI and loneliness with long-term risk of cancer.** Model 1: adjusted for age, sex, and race/ethnicity; Model 2: further adjusted for assessment center, employment, college/university degree, sun exposure time, socioeconomic status (Townsend deprivation score), smoking status, and alcohol use; Model 3: further adjusted for BMI (continuous), grip strength, family history of cancer, MAP, overall health rating, healthy diet score, healthy sleep score, and depressive mood. Rank 1: no loneliness + no isolation; Rank 2: loneliness + no isolation; Rank 3: no loneliness + isolation; Rank 4: loneliness + isolation. P-for-trend was calculated by treating rank as a continuous ordinal variable in two-sided Wald test. The HR estimates in the figure from the cause-specific Cox model (CSHR) are indicated in red, while those from the competing risk model (sHR) are shown in blue. Error bars represent 95% CI of HR. Data represent the full cohort of biologically independent sample count = 354,537 participants. CSHR Cause-Specific hazard ratio, sHR Subdistribution Hazard Ratio, HR Hazard Ratio, CI Confidence Interval, PAF Population Attributable Fraction, SI Social Isolation.

**A**

| Adjustment | HR(95%CI) | PERM |
|---|---|---|
| Minimally | 1.184 (1.138, 1.232) | |
| | 1.158 (1.110, 1.208) | |
| Socioeconomic | 1.121 (1.076, 1.167) | 34.5 |
| | 1.103 (1.056, 1.151) | 35.1 |
| Behavioral | 1.127 (1.083, 1.174) | 30.9 |
| | 1.106 (1.061, 1.152) | 33.1 |
| Physical Health | 1.173 (1.127, 1.221) | 6.1 |
| | 1.149 (1.102, 1.199) | 5.6 |
| Mental Health | 1.182 (1.135, 1.230) | 1.2 |
| | 1.157 (1.109, 1.206) | 0.8 |
| Overall Health | 1.156 (1.111, 1.203) | 15.2 |
| | 1.135 (1.089, 1.184) | 14.4 |
| Family History | 1.184 (1.137, 1.232) | 0.1 |
| | 1.158 (1.110, 1.208) | 0.1 |
| Geographical Factors | 1.182 (1.136, 1.230) | 1.0 |
| | 1.156 (1.109, 1.206) | 1.0 |
| ALL | 1.087 (1.043, 1.133) | 52.8 |
| | 1.073 (1.029, 1.120) | 53.5 |

**B**

| Adjustment | HR(95%CI) | PERM |
|---|---|---|
| Minimally | 1.276 (1.207, 1.349) | |
| | 1.265 (1.199, 1.334) | |
| Socioeconomic | 1.213 (1.146, 1.284) | 22.8 |
| | 1.205 (1.139, 1.275) | 22.6 |
| Behavioral | 1.209 (1.143, 1.278) | 24.5 |
| | 1.200 (1.137, 1.267) | 24.4 |
| Physical Health | 1.254 (1.186, 1.325) | 8.0 |
| | 1.244 (1.179, 1.313) | 7.9 |
| Mental Health | 1.274 (1.205, 1.347) | 0.7 |
| | 1.263 (1.197, 1.333) | 0.6 |
| Overall Health | 1.249 (1.182, 1.321) | 9.7 |
| | 1.241 (1.175, 1.310) | 9.1 |
| Family History | 1.275 (1.206, 1.347) | 0.6 |
| | 1.263 (1.198, 1.332) | 0.6 |
| Geographical Factors | 1.274 (1.206, 1.347) | 0.7 |
| | 1.263 (1.198, 1.332) | 0.6 |
| ALL | 1.161 (1.096, 1.229) | 41.7 |
| | 1.156 (1.091, 1.225) | 41.0 |

Cox Proportional Hazards Regression (red) — Competing Risk Regression (blue)

**Fig. 3 | Percentage of excess risk mediated by covariates for the association of social isolation with cancer incidence, in the overall population and females.** Panel (**A**) shows the PERM by covariates for SI in the overall population, and panel (**B**) shows the PERM for SI in females. The minimally adjusted model (adjusted for age, sex, and race/ethnicity) serves as the reference. HR estimates from the cause-specific Cox model (CSHR; red) and the competing risk model (sHR; blue) are shown. Error bars represent 95% CI of HR. Exact two-sided p-values in Supplementary Data 1. Data (**A**) represent the full cohort of biologically independent samples count = 354,537 participants. Data (**B**) represent the female cohort of biologically independent sample count = 190,576 participants. CSHR Cause-Specific Hazard Ratio, sHR Subdistribution Hazard Ratio, PERM Percentage of Excess Risk Attributable to Covariates, HR Hazard Ratio, CI Confidence Interval, SI Social Isolation.

of SI on breast cancer, though this mediation effect was not significant after Bonferroni adjustment (Supplementary Fig. 10, Supplementary Table 12).

In the overall population, inflammatory markers mediated the effect of SI on cancer incidence (Supplementary Fig. 9B). Neu explained 9.12% (IE0.00688 95% CI 0.00543–0.00833), white blood cells (WBC) 7.37% (IE0.00550 95% CI 0.00418–0.00681), Neu-to-Lym ratio 4.19% (IE0.00319 95% CI 0.00207–0.00432), PLT-to-Lym ratio −1.56% (IE-0.00119 95% CI −0.00183−−0.00055), and systemic inflammation 3.20% (IE0.00244 95% CI 0.00129–0.00359). In females, inflammatory markers also mediated SI's effect on overall cancer incidence (Supplementary Fig. 9C). Neu contributed 5.40% (IE 0.00725 95% CI 0.00514–0.00936), WBC 4.72% (IE 0.00630 95% CI 0.00435–0.00825), Neu-to-Lym ratio 2.42% (IE 0.00325 95% CI 0.00179–0.00470), PLT-to-Lym ratio (IE

| Variable | Count | Percent | Social Isolation HR(95%CI) | P interaction | RERI(95%CI) | Loneliness HR(95%CI) | P interaction | RERI(95%CI) |
|---|---|---|---|---|---|---|---|---|
| Overall | 354537 | 100 | 1.087(1.043 to 1.133)<br>1.073(1.029 to 1.12) | | | 0.977(0.930 to 1.026)<br>0.974(0.925 to 1.025) | | |
| Age Group | | | | 9.28e-01<br>8.18e-01 | | | 1.49e-02<br>4.02e-05 | |
| 0–49 | 84942 | 23.96 | 1.157(1.007 to 1.329)<br>1.151(1.022 to 1.298) | | | 0.792(0.674 to 0.931)<br>0.791(0.682 to 0.917) | | |
| 50–59 | 120239 | 33.91 | 1.081(1.002 to 1.167)<br>1.071(0.990 to 1.158) | | 0.018(−0.067 to 0.111)<br>0.004(−0.027 to 0.035) | 1.031(0.946 to 1.123)<br>1.029(0.950 to 1.116) | | 0.082(−0.023 to 0.199)<br>−0.003(−0.028 to 0.021) |
| 60 And Older | 149356 | 42.13 | 1.090(1.034 to 1.149)<br>1.071(1.013 to 1.132) | | −0.014(−0.092 to 0.071)<br>0.040(−0.007 to 0.088) | 0.997(0.934 to 1.064)<br>0.994(0.934 to 1.058) | | 0.055(−0.043 to 0.163)<br>−0.061(−0.109 to −0.013) |
| Sex | | | | 1.00e-02<br>2.00e-03 | | | 8.68e-02<br>6.35e-02 | |
| Female | 190576 | 53.75 | 1.161(1.096 to 1.229)<br>1.156(1.091 to 1.225) | | | 1.022(0.950 to 1.100)<br>1.020(0.956 to 1.088) | | |
| Male | 163961 | 46.25 | 1.031(0.971 to 1.095)<br>1.009(0.952 to 1.069) | | −0.100(−0.170 to −0.025)<br>0.043(0.016 to 0.070) | 0.963(0.900 to 1.029)<br>0.960(0.904 to 1.019) | | −0.081(−0.166 to 0.012)<br>0.011(−0.019 to 0.042) |
| Income | | | | 6.45e-01<br>6.40e-01 | | | 7.00e-03<br>1.11e-02 | |
| Medium | 162895 | 45.95 | 1.112(1.039 to 1.191)<br>1.107(1.035 to 1.184) | | | 0.923(0.852 to 0.999)<br>0.926(0.855 to 1.002) | | |
| High | 87548 | 24.69 | 1.011(0.857 to 1.193)<br>0.999(0.854 to 1.170) | | 0.085(−0.037 to 0.207)<br>−0.022(−0.098 to 0.061) | 0.980(0.849 to 1.132)<br>0.979(0.836 to 1.146) | | 0.064(−0.037 to 0.166)<br>0.183(0.071 to 0.307) |
| Low | 64489 | 18.19 | 1.053(0.990 to 1.120)<br>1.041(0.979 to 1.105) | | 0.046(−0.071 to 0.163)<br>0.031(−0.053 to 0.122) | 1.071(0.988 to 1.160)<br>1.068(0.981 to 1.163) | | 0.033(−0.064 to 0.131)<br>−0.088(−0.174 to 0.008) |
| Unreported | 39605 | 11.17 | 1.109(0.980 to 1.254)<br>1.093(0.959 to 1.247) | | 0.036(−0.088 to 0.176)<br>0.106(−0.029 to 0.241) | 0.831(0.716 to 0.966)<br>0.835(0.714 to 0.977) | | −0.139(−0.263 to 0.005)<br>0.074(−0.037 to 0.184) |
| Education | | | | 2.00e-01<br>2.83e-01 | | | 1.24e-01<br>1.18e-01 | |
| Not College Degree | 175971 | 49.63 | 1.101(1.045 to 1.161)<br>1.088(1.031 to 1.147) | | | 0.993(0.933 to 1.057)<br>0.994(0.937 to 1.054) | | |
| College/University Degree | 178566 | 50.37 | 1.055(0.987 to 1.128)<br>1.049(0.980 to 1.123) | | 0.003(−0.012 to 0.019)<br>0.055(−0.028 to 0.146) | 0.940(0.867 to 1.019)<br>0.940(0.867 to 1.019) | | 0.004(−0.010 to 0.018)<br>0.082(−0.021 to 0.195) |
| Employment | | | | 4.34e-01<br>4.99e-01 | | | 7.21e-04<br>2.03e-03 | |
| Employed | 211397 | 59.63 | 1.083(1.019 to 1.152)<br>1.080(1.010 to 1.154) | | | 0.888(0.823 to 0.959)<br>0.892(0.822 to 0.968) | | |
| Others | 28422 | 8.02 | 0.958(0.847 to 1.083)<br>0.942(0.837 to 1.061) | | −0.060(−0.169 to 0.063)<br>0.010(0.003 to 0.017) | 1.063(0.935 to 1.208)<br>1.067(0.931 to 1.224) | | 0.145(0.002 to 0.308)<br>0.006(−0.001 to 0.013) |
| Retired | 114718 | 32.36 | 1.111(1.044 to 1.183)<br>1.091(1.024 to 1.163) | | 0.047(−0.035 to 0.135)<br>0.003(−0.001 to 0.008) | 1.043(0.968 to 1.124)<br>1.036(0.962 to 1.117) | | 0.125(0.020 to 0.240)<br>0.008(0.004 to 0.013) |
| Smoking Satus | | | | 1.82e-01<br>1.81e-01 | | | 6.64e-01<br>7.08e-01 | |
| Current | 35339 | 9.97 | 1.081(0.992 to 1.179)<br>1.077(0.985 to 1.178) | | | 0.946(0.847 to 1.057)<br>0.950(0.840 to 1.076) | | |
| Never | 195266 | 55.08 | 1.080(1.011 to 1.153)<br>1.074(1.002 to 1.151) | | −0.030(−0.106 to 0.053)<br>−0.042(−0.071 to −0.013) | 0.966(0.894 to 1.044)<br>0.967(0.894 to 1.046) | | −0.043(−0.133 to 0.056)<br>0.004(−0.036 to 0.043) |
| Previous | 123932 | 34.96 | 1.063(0.993 to 1.139)<br>1.047(0.983 to 1.117) | | −0.036(−0.114 to 0.047)<br>−0.020(−0.042 to 0.001) | 0.983(0.908 to 1.063)<br>0.983(0.912 to 1.059) | | 0.021(−0.075 to 0.127)<br>0.013(−0.009 to 0.036) |
| Alcohol Status | | | | 8.34e-01<br>8.14e-01 | | | 2.88e-01<br>2.41e-01 | |
| Current | 329778 | 93.02 | 1.087(1.040 to 1.136)<br>1.079(1.033 to 1.127) | | | 0.964(0.915 to 1.016)<br>0.964(0.913 to 1.017) | | |
| Never | 13025 | 3.67 | 1.054(0.881 to 1.261)<br>1.032(0.864 to 1.232) | | −0.044(−0.199 to 0.142)<br>0.004(−0.002 to 0.011) | 1.063(0.843 to 1.341)<br>1.071(0.848 to 1.353) | | 0.065(−0.154 to 0.340)<br>0.008(0.004 to 0.012) |
| Previous | 11734 | 3.31 | 1.030(0.886 to 1.197)<br>1.007(0.854 to 1.189) | | 0.029(−0.112 to 0.191)<br>0.008(0.002 to 0.015) | 1.023(0.841 to 1.244)<br>1.037(0.832 to 1.294) | | 0.158(−0.046 to 0.405)<br>0.007(0.002 to 0.011) |

Lower risk — Higher risk (0.5 / 1 / 1.5)

- Cox Proportional Hazards Regression
- Competing Risk Regression

**Fig. 4 | Cancer risk associated with SI and loneliness, stratified by demographic and lifestyle factors.** HR and 95% CI (error bars) for cancer risk related to SI and loneliness, stratified by demographic and lifestyle factors. Subgroup analyses cover age, sex, income, education, employment, smoking, and alcohol use. Associations were assessed using multivariable Cox proportional hazards regression models for cause-specific hazard ratios (CSHR, red) and Fine-Gray subdistribution hazard models for subdistribution hazard ratios (sHR, blue). Multiplicative interactions were assessed using likelihood ratio tests, while additive interactions were evaluated with the Relative Excess Risk due to Interaction (RERI) derived from the Delta method. Models were adjusted for age, sex, race/ethnicity, assessment center, employment, college/university degree, sun exposure time, Townsend deprivation score, smoking status, alcohol use, BMI (continuous), grip strength, family history of cancer, MAP, overall health rating, healthy diet score, healthy sleep score, and depressive mood. Exact two-sided P values of subgroup analysis and biologically independent sample count are presented in Supplementary Data 2. CSHR Cause-Specific Hazard Ratio, sHR Subdistribution Hazard Ratio, HR Hazard Ratios, CI Confidence Intervals, RERI Relative Excess Risk due to Interaction.

−0.00093 95% CI −0.00167−−0.00018, $p = 0.0153$) and systemic inflammation (IE 0.00229 95% CI 0.00074−0.00384).

For specific cancer subtypes, Fig. 6 depicts heatmaps of inflammatory markers' mediation effects on SI-associated cancer risk, with red/green shading indicating positive/negative indirect effects (IE > 1/ < 1), color intensity reflecting significance strength, and markers "a" ($P < 0.05$, uncorrected) or "b" ($P < 0.05$ after Bonferroni correction). Significant mediation was observed for breast and lung cancers. For breast cancer, Neu, WBC and Neu-to-Lym ratio remained significant after Bonferroni correction. For lung cancer, Neu, WBC, and systemic inflammation were significant after Bonferroni correction. With Neu-to-Lym ratio and PLT to lung cancer; Neu, WBC, systemic inflammation, and PLT to bladder cancer; Neu, WBC to uterine cancer showed mediation before Bonferroni correction, but none remained significant after adjustment. In males, only WBC significantly mediated the effect of SI on bladder cancer before Bonferroni correction but not after (Supplementary Tables 13, 14). Based on these results, a Directed Acyclic Graph (DAG) was constructed to elucidate the assumed causal pathways (Supplementary Fig. 11).

## Sensitive analysis
In the sensitivity analysis, missing data for covariates were imputed, and participants who developed cancer within the first 2 years of follow-up were excluded, yielding a final sample size of 421,537 (Supplementary Table 15). Survival curves in the overall and sex-stratified analyses were consistent with those from the main analysis (Supplementary Fig. 12). The association between SI and cancer incidence also remained significant after full adjustment (CSHR1.108 95% CI 1.061–1.156; sHR1.103 95% CI 1.064–1.143) (Supplementary Fig. 13). In subgroup analyses, the increased cancer risk associated with SI remained significant and was more pronounced in females (CSHR1.143 95% CI 1.085–1.205; sHR1.145 95% CI 1.079–1.215) than in males (CSHR1.072 95% CI 1.015–1.132; sHR1.080 95% CI 1.020–1.144) (Supplementary Fig. 14). Similarly, the inverse associations between loneliness and cancer risk persisted among younger participants (aged 0–59 years; CSHR0.828 95% CI 0.718–0.956; sHR0.829 95% CI 0.715–0.962) and employed individuals (CSHR0.922 95% CI 0.860–0.988; sHR0.927 95% CI 0.867–0.992) (Supplementary Fig. 14). Specifically, associations between SI and incidence of breast, lung, uterine,

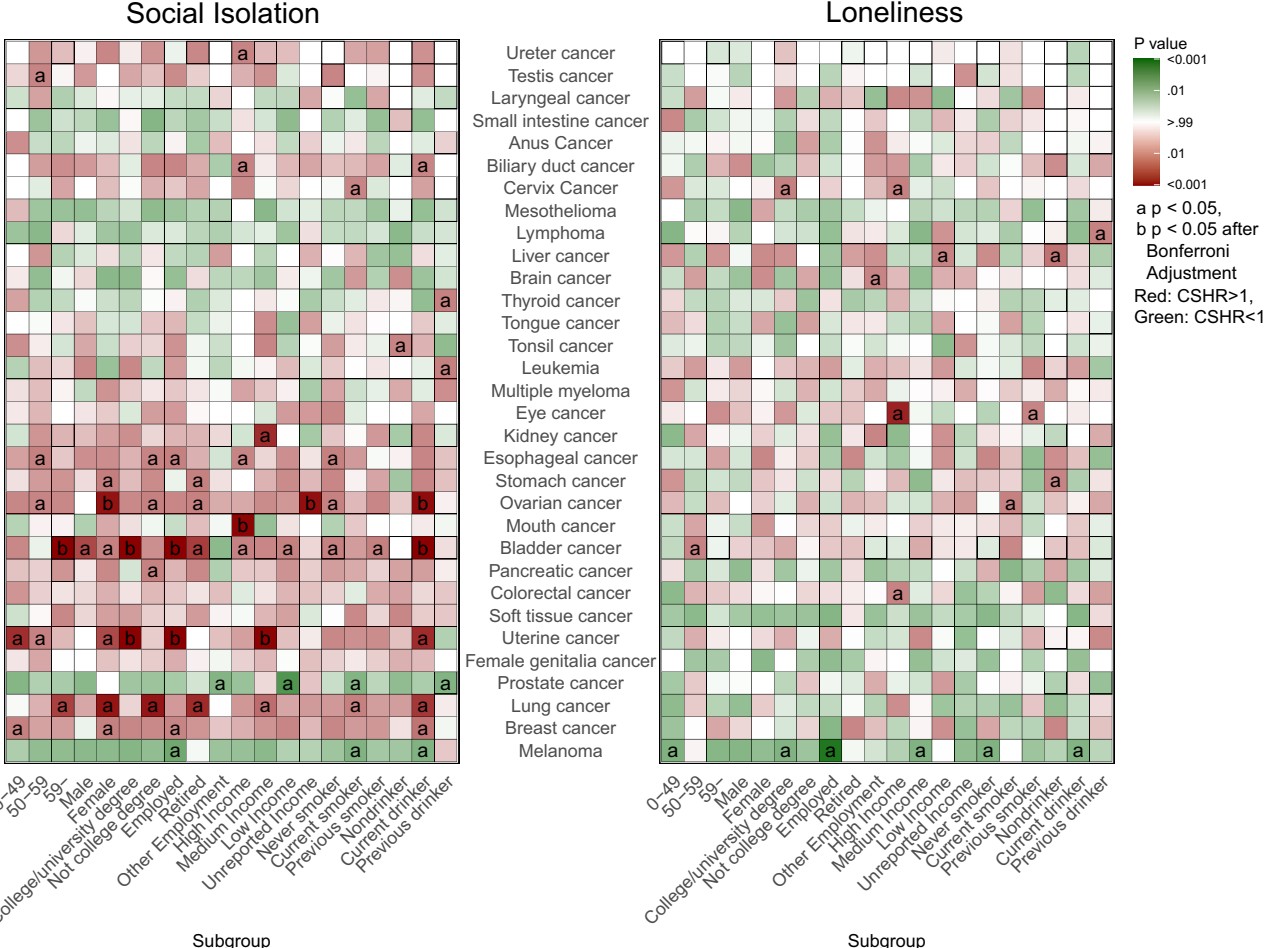

**Fig. 5 | Heatmaps of SI and loneliness associations with cancer risk across subgroups.** Heatmaps depicting the association between SI and loneliness with specific cancer risk, stratified by demographic and lifestyle subgroups. Estimates (CSHR) and P-values were derived from multivariable Cox proportional hazards regression models. The color gradient represents the direction of the effect direction, with red indicating an increase risk (CSHR > 1) and green indicating decrease risk (CSHR < 1). The depth of the color reflects the significance of the association, with darker shades indicating smaller P-values. Significant associations are marked with "a" for two-sided P < 0.05 and "b" for two-sided P < 0.05 after Bonferroni correction. Exact p-values, CSHR, and subgroup-specific biologically independent sample count are in Supplementary Data 3. CSHR Cause-Specific Hazard Ratio, SI Social Isolation.

ovarian, and bladder cancer in females, as well as bladder cancer in males, remained apparent (Supplementary Figs. 15, 16).

## Discussion

In this UK Biobank study, we found that SI was significantly associated with higher cancer risk, whereas loneliness was not significantly associated with cancer risk. A strong interaction was observed between sex and SI, with the effect being more pronounced in female participants, who exhibited a higher overall cancer risk and a greater number of cancer sites compared to males. The association between SI and cancer risk is mediated by inflammatory markers and, in the case of uterine cancer in females, by SHBG acting as a negative mediator. Our findings may provide insights into the effects of SI and loneliness on health, and highlight the need for incorporate SI into cancer prevention strategies, particularly given its stronger effect on females.

To our knowledge, this is the first comprehensive study to examine the influence of SI and loneliness on various cancer types, revealing significant differences between sexes. Our study found that SI exposure in female participants was associated with increased risks of lung, breast, uterine, bladder, and ovarian cancers. In male participants, SI associated with an increased risk of bladder cancer. These findings are consistent with prior studies. For instance, the UK Biobank cohort reported that SI was associated with a 32% higher risk of mortality due to neoplasms[4]. Previous research also supports the association between SI and elevated risks of breast[16], ovarian[38],

and lung cancers in female[39]. Our study extends these findings by demonstrating specific associations between SI and individual cancer types, highlighting the need to consider SI as a significant factor in cancer risk assessments. Loneliness showed no significant association with cancer risk in the general population, consistent with previous study[40].

Notably, subgroup analyses revealed significant interactions between loneliness and cancer risk by age, sex, income, education, and employment. Among low-income, non-college-educated, and unemployed individuals, additive interaction effects indicated that loneliness had a stronger contribution on cancer risk in these groups. These findings underscore heterogeneity that may explain the null overall effect, for the aggregate analyses likely masked subgroup-specific signals. Conversely, inverse associations emerged in younger and employed participants, as confirmed by competing-risk models and sensitive analysis. These patterns align with evolving views of loneliness as context-dependent[30]: in resilient groups like employed or middle-income individuals, of whom have high autonomy and social capital, transient loneliness may foster adaptive behaviors, such as forging quality connections[41] or prioritizing well-being over unhealthy norms[42,43]. Such nuances highlight opportunities for precision interventions targeting vulnerable subgroups and refining cancer prevention strategies, though they require cautious interpretation and further study to confirm mechanisms.

Our study shows that while SI and loneliness may be conceptually related, they play distinct and independent roles in cancer incidence. SI has a

**Fig. 6 | Mediation effects of inflammatory markers on specific cancer risk with SI exposure.** Heatmaps depicting the IE of inflammatory markers in mediating the relationship between SI and specific cancer risk. Mediation analyses were performed using the counterfactual framework, with significance assessed via the bias-corrected bootstrap method (resampling = 1000). The color gradient represents the direction of the mediation effect, with red indicating a positive indirect effect (IE > 1) and green indicating a negative indirect effect (IE < 1). The depth of the color reflects the significance of the association, with darker shades indicating smaller *P*-values. Significant associations are marked with "a" for two-sided *P* < 0.05 and "b" for two-sided *P* < 0.05 after Bonferroni correction. Exact *p*-values, estimation, and subgroup biologically independent sample count are in Supplementary Data 4. IE Indirect Effect, Lym Lymphocytes, Mono Monocytes, Neu Neutrophils, PLT Platelets, WBC White Blood Cells, CRP C-reactive Protein, Lym-mono Ratio Lymphocyte to Monocyte Ratio, Neu-lym ratio Neutrophil to Lymphocyte Ratio, PLT-lym Ratio Platelet to Lymphocyte Ratio, SI Social Isolation.

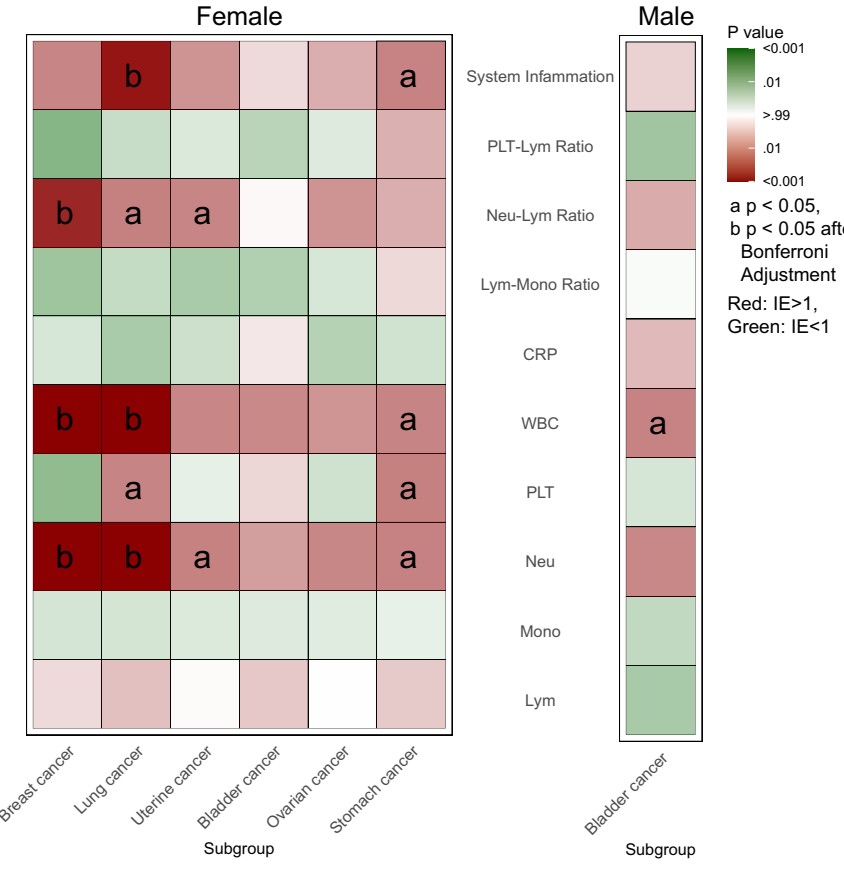

greater influence on cancer prognosis than loneliness, implying their different contributions in cancer risk[4]. SI, as an objective state of reduced social contact, directly influences health behaviors and biological processes, including stress[44], pro-inflammatory responses[45–47], and altered cellular pathways[48], all of which contribute to cancer development. In contrast, loneliness is a subjective feeling with more complex and indirect effects[3]. Additionally, individuals with SI may face heightened cancer risk due to reduced screening and unhealthy lifestyles[2], as evidenced by our PERM analysis, which indicates that lifestyle mediates the SI-cancer risk relationship. A meta-analysis of 90 cohort studies found that SI is significantly associated with higher all-cause and cancer-specific mortality than loneliness[49], supporting our findings and the need for further research on mediators in the SI-cancer link. Our study identified a significant interaction between SI and sex, with a more pronounced effect on females. SI notably increased the risk of female-specific cancers, such as uterine, breast, and ovarian cancer, underscoring females' heightened susceptibility. We assessed whether hormone levels and menstrual status mediated the association between SI and overall cancer risk in females, but found no significant mediation. However, at the individual cancer level, SHBG acts as a suppressor for uterine cancer in females. These results suggest further exploration of other potential hormone mediators. Additionally, sex differences in stress response are well-documented: with females often adopting affiliative coping strategies and males exhibiting more competitive responses[50]. These differences in stress response may contribute to females' greater vulnerability to SI, as observed in our study. Future research is needed to explore potential neurophysiological and psychological mechanisms underlying these sex-specific effects.

Pro-inflammatory mechanisms have been shown to serve as a pathway through which SI increases the risk of all-cause mortality and chronic diseases[51], which may also contribute to cancer development. Stressors like SI can activate the HPA axis and the SNS, leading to dysregulation of various neuroendocrine hormones, particularly catecholamines and cortisol[52,53]. This hormonal imbalance can directly contribute to cancer progression by

inducing DNA damage and enhancing p53 degradation. More importantly, they can alter the tumor microenvironment by disrupting immune function and promoting chronic inflammation, which impairs immune surveillance[26]. In our study, WBC count and Neu were found to mediate the effect of SI on cancer risk in the overall population and females. Specifically, in female participants with lung cancer, systemic inflammation, platelet levels, and the neutrophil-to-lymphocyte ratio all showed significant mediation effects. These findings align with previous research, confirming the pivotal role of inflammation in SI-related disease mechanisms[26]. However, the specific immune pathways mediating the effect of SI on cancer incidence remain unclear, necessitating the need for further investigation.

From an applied perspective, this large-scale study exploring the effect of SI, loneliness on cancer risk, our findings broaden the understanding of the health consequences associated with SI and underscore the need for greater attention to both SI and loneliness. These insights could inform public health policies and contribute to more effective cancer prevention strategies. We recommend developing assessment tools and preventive interventions to identify high-risk individuals and reduce cancer incidence associated with SI and loneliness. Specifically, at the individual level, interventions could include promoting social participation and mindfulness training, which has been shown to mitigate the negative effects of SI[31]. Furthermore, at the community level, social support systems can help isolated and lonely individuals reintegrate into society and prevent disease. Meanwhile, at the societal level, public health strategies should reduce the consequence of SDOH and provide comprehensive medical screening. Additionally, digital interventions, such as online platforms, can offer social support to those who are isolated. To reduce cancer risk, addressing SI and loneliness directly is crucial, as well as developing effective preventive strategies to mitigate the effects of potential mediating factors.

This study has several strengths. First, the large sample size, long-term follow-up, and prospective design enhance statistical power, increase representativeness and external validity, allow accurate assessment of long-term risks and outcomes, and establish clear temporal relationships between

variables, strengthening causal inferences. Second, meticulous control of multiple covariates and the use of well-established methodologies minimizes confounding factors and ensure scientific rigor and reliability in data collection and analysis. Third, the comprehensive and well-structured design systematically evaluates the association of various factors on outcomes, and we explored potential pathological and biological mechanisms underlying the observed associations. Lastly, strict P-value corrections enhance the robustness of the findings and reduce the risk of false positives. Collectively, these strengths elevate the scientific rigor and reliability of the study.

Despite these strengths, our study has several limitations. First, selection bias may have arisen from lower participation among those with greater SI or loneliness for their less willing to participate, as well as the predominance of middle-aged to older adults of European descent in the UK Biobank cohort; thus, results should be generalized cautiously to younger or non-European populations. Non-random exclusion due to missing data further introduced imbalances in low socioeconomic status, psychological distress, and assessment center representation (Supplementary Tables 16–18). However, standardized mean differences (SMD) for key exposures (SI and loneliness) were negligible (<0.1), supporting the robustness of primary findings. Second, SI and loneliness were assessed via brief, self-reported questions adapted from validated instruments and used in prior studies, but these focused solely on in-person contacts and may underestimate virtual interactions, potentially reducing measurement precision. Third, assessments were limited to baseline, without longitudinal repeats in the full cohort. Although repeats were available for a subset, we prioritized statistical power and follow-up duration by excluding them due to high missingness and selection bias; supplementary analyses nonetheless confirmed temporal stability of SI and loneliness in this subset (Supplementary Tables 19, 20). Additionally, our mediation models did not fully disentangle mediators from confounders and omitted other candidates, such as detailed behavioral or physiological markers. Finally, as an observational study, residual confounding and reverse causation cannot be fully excluded, despite prospective design and extensive covariate adjustment.

## Conclusions

In conclusion, our findings suggest that social isolated people are at a moderately increased risk of cancer incident. There is a significant difference across various populations, with females being particularly affected and facing a higher risk. Interventional studies are required before policy recommendations are proposed.

## Data availability

UK Biobank data are available through application to the database (https://www.ukbiobank.ac.uk/)[50]. Researchers registered with UK Biobank can apply for access to the database by completing an application, which includes a summary of the research plan, data fields required, any new data or variables that will be generated, and payment to cover the incremental costs of servicing an application. The source data for Fig. 3 is in Supplementary Data 1, the source data for Fig. 4 is in Supplementary Data 2, the source data for Fig. 5 is in Supplementary Data 3, the source data for Fig. 6 is in Supplementary Data 4, the source data for Supplementary Fig. 3 is in Supplementary Data 5, the source data for Supplementary Fig. 4 is in Supplementary Data 6, the source data for Supplementary Fig. 5 is in Supplementary Data 7, the source data for Supplementary Fig. 10 is in Supplementary Data 8, the source data for Supplementary Fig. 14 is in Supplementary Data 9.

## Code availability

The analysis code for this study has been deposited in the Zenodo repository and is available at https://doi.org/10.5281/zenodo.18197478[37]. The associated GitHub address is https://github.com/DrCheng769/R-Code-of-SI-and-Loneliness-in-Cancer-Risk.

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

## Author contributions

All authors participated in designing the study, generating hypotheses, interpreting the data, and critically reviewing the report. C.J.H., W.R.C., and F.Y. were primarily responsible for writing the article. C.J.H., with help from W.R.C. and F.Y., did the data analyses. C.J.H., W.R.C., and F.Y. accessed and verified the data. Y.S.J., L.H.R., C.B., and C.Q. contributed to data collection and analysis, and reviewed the manuscript. C.J.H. and F.Y. contributed to the study design, project administration, and supervised the work. H.J.X. and L.W.H. provided funding acquisition and supervised the project. All authors confirm they had access to all the data in the study and accept responsibility for the decision to submit for publication. The corresponding authors had full access to the data in the study and take responsibility for the integrity of the data and the accuracy of the data analysis.

## Competing interests

The authors declare no competing interests.
