## [Transparent Peer Review file · Communications Medicine]

A study of the associations between social isolation and loneliness with sex-specific cancer risk in the UK Biobank

Corresponding Author: Dr Jiahao Cheng

Version 0:

Reviewer comments:

Reviewer #1

(Remarks to the Author)

This study explores an important and understudied question about the relationship between social isolation (SI), loneliness, and cancer risk. A major strength of the study is the use of a large prospective cohort, which enabled analysis across multiple cancer subtypes. However, the manuscript has several notable limitations. First, the rationale for investigating the association between social isolation, loneliness, and cancer risk is not clearly articulated. Given the large number of analyses performed, the paper would benefit from clearly stated, a priori hypotheses to guide the reader and justify the analytic approach. Without this, the work risks appearing exploratory rather than hypothesis driven. Additionally, the presentation of results is difficult to follow due to the sheer volume of analyses and limited synthesis of key findings. As outlined below, several specific issues should be addressed to improve the clarity, focus, and interpretability of the manuscript.

1. Abstract (Lines 83-84), this sentence is contradictory: "Social isolation (SI) and loneliness are significant factors in cancer development, however, their specific relationships with cancer incidence remain understudied."
2. The Introduction lacks a clear rationale for why social isolation and loneliness may contribute to increased cancer risk. The authors should provide more detail on the potential biological or behavioral pathways through which these factors could influence cancer development. This is particularly important for justifying the mediation analysis.
3. Introduction (Lines 101-103): Need citations for this statement: "SI and loneliness are widespread, with an estimated 1 in 4 older people experiencing social isolation and between 5 and 15 percent of adolescents experiencing loneliness."
4. The authors should reference the flow chart in the description of the Study Population in the Methods Section. Along this same point, not all the supplementary figures appear to be referenced in the text. For example, I did not see a reference to eFigure1.
5. Statistical Analysis (Line 333): "Individuals reporting SI or loneliness at baseline were classified as the exposure and sex group." This is confusing. Shouldn't it just say "classified as the exposure"?
6. There are limitations to using the Baron and Kenny approach to evaluate mediation. Why not use a more advanced mediation method for this analysis?
7. The figures, particularly the heatmaps, are difficult to interpret and would benefit from clearer presentation. Figure 3 lacks column labels, which makes it hard to understand the structure and meaning of the data. In Figure 4, the interpretation of the green and red shading is unclear—do the colors represent the magnitude of effect estimates, statistical significance (e.g., p-values), or directionality of associations? This needs to be explicitly stated in the figure legend. Similarly, Figure 5 is difficult to follow, and the legend does not adequately explain the content. It is also unclear what distinguishes panel a from panel b. Overall, the figures would benefit from clearer labeling, more informative legends, and guidance in the text to aid interpretation.
8. Results (Lines 150-152): What is the hypothesis underlying the observed inverse association between loneliness and cancer risk among younger and employed adults? These findings are not addressed in the Discussion section, leaving the

reader without context or interpretation. Without a clearly articulated rationale for conducting these subgroup analyses, the results risk appearing spurious or data-driven rather than hypothesis-driven. The authors should either provide a plausible explanation for these findings or acknowledge the potential for chance associations.

9. Results (Lines 161-163): "Figure 4 showed the relationship between various population characteristics and cancer risk, showing that SI significantly affects cancer risk across several groups. SI was significantly associated with an increased risk of five cancer after adjustment." – Can the authors be more specific here and state which cancer types they are referring to. It is very hard for the reader to keep track of what is significant and what is not.

10. Results (Line 165): What does "without clear income" mean?

11. Not every abbreviation in the figures (for example, eFigure 9) is defined in the legend.

12. Results (Line 154): "After adjusting for age, sex, and ethnicity, SI was associated with higher cancer risk in both sexes (eFigure 4, eFigure 5)." – No need to say adjusting for sex in this sentence as this is a stratifying variable here.

13. Discussion (Lines 200-201): "The association between SI and cancer risk is partly mediated by hormonal status and inflammatory cells." Where are the results that support hormonal status as a significant mediator? Based on eFigure 9, hormonal status does not appear to be a significant mediator. Additionally, this statement appears to contradict the statement on Lines 228-230: "We assessed whether hormone levels and menstrual status mediated this effect, but found no significant mediation, indicating a need to investigate other potential mediators."

14. Discussion (Lines 230-233): "Additionally, sex differences in stress response are well-documented: females, with a more developed neocortex, tend to adopt affiliative coping strategies, while males, relying more on sub-cortical structures, exhibit more competitive and combative responses^{23,24}. These neurophysiological differences may partly explain female's greater vulnerability to SI." This seems like a stretch, as the data in this study tell us nothing about neurophysiological differences between sexes.

15. A major limitation, as noted by the authors, is the reliance on a single measure of SI and loneliness at baseline. The authors state on Lines 277-278: "Although repeated measurements were available for part of the cohort, we chose not to include them to maximize statistical power and follow-up duration." Given this, why not examine how stable these measures are over time within the subset of your sample that has repeated measures. If it can be shown that these measures don't change much over time, it would add validity to the study findings.

Reviewer #2

(Remarks to the Author)

This large UK Biobank study found that social isolation, but not loneliness, was associated with an 8% increased risk of cancer, with a stronger effect observed in women and for several cancer types. The associations were largely explained by socioeconomic, behavioral, and inflammatory factors, highlighting the importance of targeted prevention strategies for socially isolated individuals. Overall interesting paper. I have a few suggestions:

1. in abstract, it would be great if the author could clearly define what is meant by "social isolation" and "loneliness" within the context of their study.
2. in background, it would be helpful for the authors to discuss the potential mechanisms by which social isolation and loneliness may affect the incidence of various cancer.
3. the description of the study population needs further clarification. The author stated "a prospective cohort study that recruited 502,258 participants (18.1% of the eligible population) in the UK between January 1, 2006, and December 31, 2010." Could the authors clarify what is meant by "18.1% of the eligible population"?
4. it seems the adjusted models appear to include many variables, some are highly correlated. How did the author address the multicollinearity issue?
5. ignoring competing events in survival analysis may lead to biased results, the results from the competing risk model should be presented as the main findings rather than as a sensitivity analysis.
6. the author used complete cases only. It would be better to compare characteristics between included and excluded participants and discuss any potential bias arising from the exclusion of cases with missing data.
7. in the main table, there is p for trend reported under the joint effect analysis. While it makes sense to consider an internal order between "No isolation in no loneliness" and "Isolation in loneliness," the rationale for ranking "No isolation in loneliness" versus "Isolation in no loneliness" is unclear. Could the authors clarify their approach here?

Version 1:

Reviewer comments:

Reviewer #1

(Remarks to the Author)

The authors have addressed all my comments and concerns.

Reviewer #2

(Remarks to the Author)

I appreciate the author's comprehensive reply. They have fully addressed my concerns with clear revisions, additional analyses.

**Reviewers' comments:**

**Reviewer #1 (Remarks to the Author):**

**This study explores an important and understudied question about the relationship between social isolation (SI),**
**loneliness, and cancer risk. A major strength of the study is the use of a large prospective cohort, which enabled**
**analysis across multiple cancer subtypes. However, the manuscript has several notable limitations. First, the**
**rationale for investigating the association between social isolation, loneliness, and cancer risk is not clearly**
**articulated. Given the large number of analyses performed, the paper would benefit from clearly stated, a priori**
**hypotheses to guide the reader and justify the analytic approach. Without this, the work risks appearing exploratory**
**rather than hypothesis driven. Additionally, the presentation of results is difficult to follow due to the sheer volume**
**of analyses and limited synthesis of key findings. As outlined below, several specific issues should be addressed to**
**improve the clarity, focus, and interpretability of the manuscript.**

We appreciate the reviewers' insightful feedback, which has substantially strengthened our manuscript. We have carefully
considered all the points raised by the reviewers and have made revisions accordingly. Major highlights include:

- (1) Refined abstract with precise SI/loneliness definitions (objective vs. subjective)
(2) Replenish prevalence citations (20–25% older adults; 5–15% adolescents)
(3) Added mechanistic pathways(HPA/SNS stress, behavioral risks, protective effects) in introduction
(4) Hypothesizing loneliness-linked risk reduction in younger/employed via adaptive coping
(5) Clarified methods by correcting participation rate to 5.4% (UK Biobank data)
(2) Referencing all figures/eTables, and added full abbreviations
(3) Optimized figure labels/legends (clearer heatmaps/forests for Figures 4–6)
(4) Addressed phrasing (e.g., "unreported income"; naming "five cancers": ovarian, uterine, bladder, lung, mouth)
(5) Resolved multicollinearity (VIF<2 via MAP/BMI adjustments)
(6) Integrated competing risks as main results (CSHR/sHR dual-reporting, CIFs)
(7) Assessed missing data bias via included/excluded comparisons (SMD<0.3 for exposures; eTables 9–11)
(8) Added repeated-measure stability (>92% agreement in subset)
(9) Refined joint effects with explicit ranks 1–4 (no SI/no loneliness to SI+loneliness)
(10) Enhanced mediation to advanced causal methods (VanderWeele)
(15) Toned sex-stress claims to behavioral evidence

These revisions boost rigor, transparency, and focus, amplifying implications for female-targeted cancer prevention, in line
with Nature-series standards. Detailed point-by-point responses follow.

**Comment 1: Abstract (Lines 83-84), this sentence is contradictory: "Social isolation (SI) and loneliness are**
**significant factors in cancer development, however, their specific relationships with cancer incidence remain**
**understudied."**

Reply: We thank the reviewer for this keen observation on the abstract's phrasing, which highlights an opportunity to sharpen our
distinction between prognostic and incidence evidence—your insight elevates its precision. We agree the original implied broad
"cancer development" roles, potentially misleading amid strong mortality data but limited incidence links.

To resolve this, we have revised the opening sentence as follows in the abstract section. Original: "Social isolation (SI) and
loneliness are significant factors in cancer development, however, their specific relationships with cancer incidence remain
understudied."; Revised: "**Both SI and loneliness are established risk factors for poor cancer prognosis, including increased**
**mortality, yet their associations with cancer incidence remain understudied."**

This revision clearly delineates that existing evidence primarily focuses on the prognostic impact of SI and loneliness,
whereas our study centers on their associations with cancer incidence. At the same time, this framing highlights the novelty of our
work in addressing this understudied aspect, thereby strengthening the overall rationale.

We believe this revision enhances the abstract's clarity and precision. Again, we are grateful for your thoughtful feedback,
which has significantly improved the quality of our manuscript. We hope this revision adequately addresses your concern, and we
remain open to further suggestions to enhance it. Thank you again for your constructive input.

Changes in the text:

Line 43-44, the section of "Abstract".

**Comment 2: The Introduction lacks a clear rationale for why social isolation and loneliness may contribute to**
**increased cancer risk. The authors should provide more detail on the potential biological or behavioral pathways**
**through which these factors could influence cancer development. This is particularly important for justifying the**
**mediation analysis.**

Reply: We sincerely thank the you for highlighting this critical point, which is essential for clarifying the mechanistic
underpinnings of our study and justifying our mediation analyses. We acknowledge that our original Introduction did not
sufficiently elaborate on the biological and behavioral pathways linking SI and loneliness to cancer risk, and we are grateful for
your guidance in addressing this key gap to enhance the manuscript's rigor and clarity.

To respond, we have revised the Introduction by expanding the mechanistic discussion into two primary pathways of
physiological alterations and behavioral changes. The physiological pathway involves stress responses that may impair immune
function and promote tumor growth, while the behavioral pathway focuses on lifestyle factors that increase cancer risk. Moreover,
we incorporated potential protective nuances for a balanced perspective. These additions not only clarify how SI and loneliness
may elevate cancer risk but also directly support our subsequent mediation explorations of inflammation and hormones.

The revised is now in the introduction. **Revision: "Plausible mechanisms suggest that SI and loneliness may increase**
**cancer risk primarily through physiological alterations and behavioral changes^{4,18,19}. Physiologically, SI and loneliness act**
**as chronic stressors²⁰, activating the hypothalamic-pituitary-adrenal (HPA) axis and leading to sympathetic nervous**
**system (SNS) dysregulation²¹. This triggers hormonal imbalances, such as elevated cortisol levels, and chronic**
**inflammation²²⁻²⁴, impairing DNA repair, promoting oncogene activation, and facilitating tumor initiation and**
**progression^{22,25,26}. Behaviorally, affected individuals often adopt unhealthy habits, such as increased smoking, excessive**
**alcohol consumption, reduced physical activity, poor sleep, and suboptimal diet, all of which are established risk factors**
**for cancers like lung, colorectal, and breast cancer^{4,27}.**

**However, SI and loneliness may also confer potential protective effects in certain contexts. Reduced social interactions**
**could limit exposure to environmental carcinogens, such as second-hand smoke, alcohol, or infectious agents like HPV,**
**HBV, and H. pylori^{12,28}. Moreover, loneliness's subjective distress might spur adaptive re-engagement, mitigating long-term**
**isolation^{29,30,31}. Despite these complexities, there is still a lack of comprehensive large-scale cohort studies that can**
**systematically and thoroughly demonstrate the relationship between loneliness, SI, and cancer incidence."** These balanced
yet focused revisions (highlighted in yellow in the tracked-changes version) are grounded in seminal meta-analyses and cohort
studies, ensuring robustness while enhancing readability for a broad audience. By delineating these pathways, the changes forge a
stronger logical bridge to our mediation results and underscore opportunities for targeted cancer prevention strategies.

We are deeply grateful for your expertise, which refined our manuscript to better illuminate the public health implications of
our work. We welcome any additional suggestions to further polish the manuscript.

Changes in the text:

Line 67-79, the section of "Introduction".

**Comment 3. Introduction (Lines 101-103): Need citations for this statement: "SI and loneliness are widespread, with**
**an estimated 1 in 4 older people experiencing social isolation and between 5 and 15 percent of adolescents**
**experiencing loneliness."**

Reply: We sincerely thank you for your perceptive comments regarding the need for citations to support the prevalence statement
on SI and loneliness. Your keen eye for evidentiary gaps has greatly helped us strengthen the scientific rigor and clarity of our
work. We deeply appreciate your expertise and guidance in elevating our manuscript.

We fully acknowledge that our original submission: "SI and loneliness are widespread, with approximately 20%–25% of
community-dwelling older adults experiencing social isolation and approximately 5 and 15 percent of adolescents experiencing
loneliness" lacked supporting references. We agree that robust epidemiological evidence is essential to substantiate such claims,
especially given the public health significance of SI and loneliness.

In response, we have integrated authoritative citations for the prevalence estimates. For older adults, we now cite Lyu, C. et
al. (2024)¹, which reports that approximately **20%–25%** of community-dwelling older adults experience social isolation. For
adolescents, we reference three studies to support the **5%–15%** loneliness prevalence range, reflecting its variability across
diverse populations and measurement approaches: Firstly, Surkalim, D. L. et al. (2022)² report a pooled prevalence of loneliness
ranging from **9.2%** (95% CI: 6.8%–12.4%) in South-East Asia to **14.4%** (95% CI: 12.2%–17.1%) in the Eastern Mediterranean,
highlighting regional and cultural differences that contribute to the range's breadth. Secondly, Zoellner, F. et al. (2025)³ indicate
that loneliness rates among adolescents remained stable at **9.5%–15.5%** from 2003 to 2017. Thirdly, Hosozawa, M. et al. (2022)⁴
find that approximately **5%** of adolescents report frequent or chronic loneliness, capturing the lower end of the range for specific
subgroups. In summary, we chose the 5%–15% range to reflect the variability in prevalence due to differences in geographic
regions, cultural and socioeconomic factors, measurement tools, and time periods.

To clarify variability and preempt reader queries, we revised the sentence as follows: " SI and loneliness are widespread,
with approximately 20%–25% of community-dwelling older adults experiencing SI⁵, and approximately 5% – 15% of adolescents
experiencing loneliness⁶⁻⁸, varying by population and measurement approaches. As issues affecting people of all ages, particularly
older adults and adolescents, SI and loneliness are increasingly recognized as priority public health problems⁹."

These concise yet nuanced revisions, which are highlighted in yellow in the tracked-changes version, can enhance
transparency and evidentiary foundation directly addressing your concern. We sincerely thank you again for your acute feedback,
which has not only resolved this limitation but inspired a more compelling narrative on SI and loneliness as modifiable risks.

Changes in the text:

Line 62-65, the section of "Introduction".

Reference

1 Lyu, C., Siu, K., Xu, I., Osman, I. & Zhong, J. Social Isolation Changes and Long-Term Outcomes Among Older
Adults. *JAMA Netw Open* 7, e2424519 (2024). <https://doi.org/10.1001/jamanetworkopen.2024.24519>

2 Surkalim, D. L. et al. The prevalence of loneliness across 113 countries: systematic review and meta-analysis. *BMJ*
376, e067068 (2022). <https://doi.org/10.1136/bmj-2021-067068>

3 Zoellner, F. et al. Two decades of loneliness among children and adolescents: longitudinal trends, risks and resources -
Results from the German BELLA and COPSY studies. *Eur Child Adolesc Psychiatry* (2025).
<https://doi.org/10.1007/s00787-025-02779-6>

4 Hosozawa, M. et al. Predictors of chronic loneliness during adolescence: a population-based cohort study. *Child*
*Adolesc Psychiatry Ment Health* 16, 107 (2022). <https://doi.org/10.1186/s13034-022-00545-z>

**Comment 4. The authors should reference the flow chart in the description of the Study Population in the Methods**

**Section. Along this same point, not all the supplementary figures appear to be referenced in the text. For example, I**
**did not see a reference to eFigure1.**

Reply: We thank the reviewer for highlighting this important point, which is crucial for enhancing the manuscript's clarity and
accessibility. We deeply appreciate the reviewer's meticulous reading and valuable guidance. We fully agree with the comment and
have taken it very seriously.

In response, we conducted a thorough, systematic check of the entire manuscript to ensure that every tables, figures and all
the supplementary items (eTables and eFigures) is now appropriately cited in the main text. We apologize for the oversight in our
initial submission. The specific changes made are detailed below for the reviewer's convenience:

The study flow chart (**Figure1**) is now explicitly referenced in the Results Section, Characteristics of Patient(line 89):
"...were assessed in a median follow-up time of 11.60 years [IQR 8.40–12.72] (Figure1)."

**We also revised all the eTable and eFigure, here is the details:**

**eFigure1:**Referenced in Results Section, The Association Between SI, Loneliness and Cancer Risk in Strata Analysis (line
98, 120, and 121): "In the joint effect, as the degree of isolation and loneliness increases, the risk of cancer incidence rises (Table2,
eFigure1). "; "cohort (females: CIF P<0.001, KM P<0.001; males: CIF P=0.005, KM P<0.001)(eFigure1C, E)"; " (females: CIF
P=0.021, KM P=0.013; males: CIF P=0.028, KM P=0.078)(eFigure1D, F)."

**eFigure2:** Referenced in the Results Section, The Association Between SI, Loneliness and Cancer Risk in Strata Analysis
(line 115 and 116): ".. showed a significant association with cancer risk after adjustment (eFigure2B,C). "; "was not significant
even with minimal adjustment (eFigure2A,D)."

**eFigure3-4:** Referenced in the Discussion Section (line 141): "...higher cancer risk in both male and female participants
(eFigure3-4)."

**eFigure5:** Referenced in Results Section, SI and Loneliness Increase the Specific Cancer Risk in Various Population (line
142, 145): "...,SI increased the risk of bladder cancer (CSHR1.50 95%CI 1.12-2.01)in male participants (eFigure5A)."; "In
females, SI was linked to higher risks of several cancers... (eFigure5B)."; "No site-specific risks emerged for loneliness exposure
after adjustment (eFigure5)."

**eFigure6 and eFigure7:** Referenced in Results Section, SI and Loneliness Increase the Specific Cancer Risk in Various
Population (line 146): "Histological subtype analyses revealed no significant SI associations in either sex (eFigure6-7)."

**eFigure8:** Referenced in Results Section, Mediation Analysis (line 166, 169 and 173): "The mediation analysis revealed that
inflammatory markers and specific hormonal factors partly mediated the association between SI and cancer incidence. Female sex
hormones and menstrual status had no significant impact on overall cancer risk (eFigure8A)."; "In the overall population,
inflammatory markers mediated the effect of SI on cancer incidence (eFigure8B)."; "In females, inflammatory markers also
mediated SI's effect on overall cancer incidence (eFigure8C)."

**eFigure9:** Referenced in Results Section, Mediation Analysis (line 168): "SHBG significantly mediated the association
between SI and uterine cancer in female participants after adjustment... (eFigure9, eTable5)."

**eFigure10:** Referenced in Results Section, Mediation Analysis (line 184): "Based on these results, a Directed Acyclic Graph
(DAG) was constructed to elucidate the assumed causal pathways (eFigure10)."

**eFigure11:** Referenced in Results Section, Sensitive Analysis (line 189): "Survival curves in the overall and sex-stratified
analyses were consistent with those from the main analysis (eFigure11)."

**eFigure12:** Referenced in Results Section, Sensitive Analysis (line 191): "The association between SI and cancer incidence
also remained significant after full adjustment (CSHR1.10895%CI 1.061-1.156; sHR1.103 95%CI 1.064-1.143) (eFigure12)."

**eFigure13:** Referenced in Results Section, Sensitive Analysis (line 193): "In subgroup analyses, the increased cancer risk
associated with SI remained significant and was more pronounced in females... (eFigure13)."

**eFigure14 and eFigure15:** Referenced in Results Section, Sensitive Analysis (line 197): "..., as well as bladder cancer in
males, remained apparent (eFigures14-15)."

**eFigure16:** Referenced in Methods Section, Statistics Analysis (line 364): "No violations of the proportional hazards

assumption were observed for the exposures using Schoenfeld residuals (eFigure16)."

**eTable1:** Referenced in Results Section, Characteristics of Patient (line 95): "...and15,942(4.5%) had an exposure of
loneliness (Table1, eTable1)."

**eTable2:** Referenced in Results Section, The Association Between SI, Loneliness and Cancer Risk in Strata Analysis (line
130): "...(CSHR1.087 95%CI 1.040-1.136;sHR1.079 95%CI 1.033-1.127) (Figure4, eTable2)."

**eTable3:** Referenced in Results Section, The Association Between SI, Loneliness and Cancer Risk in Strata Analysis (line
137): "...all showing stable reduced cancer risk (Figure4, eTable3)."

**eTable4:** Referenced in Results Section, SI and Loneliness Increase the Specific Cancer Risk in Various Population (line
161): "...after correction, these associations were not statistically significant (eTable4)."

**eTable5:** Referenced in Results Section, Mediation Analysis (line 168): "SHBG mediated the effect of SI on breast cancer,
though this mediation effect was not significant after Bonferroni adjustment (eFigure9, eTable5)."

**eTable6 and eTable7:** Referenced in Results Section, Mediation Analysis (line 183): "...only WBC significantly mediated
the effect of SI on bladder cancer before Bonferroni correction but not after (eTable6-7)."

**eTable8:** Referenced in Results Section, Sensitive Analysis (line 188): "...excluding participants who developed cancer
within the first 2 years of follow-up were excluded, yielding a final sample size of 421,537 (eTable8)."

**eTable9, eTable10, and eTable11:** Referenced in Discussion Section, Study Strengths and Limitation (line 277):
"...psychological distress, and assessment centre representation (eTables9-11)."

**eTable12 and eTable13:** Referenced in Discussion Section, Study Strengths and Limitation (line 283): "...supplementary
analyses nonetheless confirmed temporal stability of SI and loneliness in this subset (eTables12-13)."

**eTable14 and eTable15:** Referenced in Methods Section, Outcome Ascertainment (line 323): "...mitigating overestimation
of risks relative to KM estimates (eTable14-15)."

**eTable16:** Referenced in Methods Section, Outcome Ascertainment (line 324): "...To ensure analytical robustness, we
focused on cancer types with more than 100 cases in the individual cancer analysis (eTable16)."

**eTable17 and eTable18:** Referenced in Methods Section, Mediator Ascertainment (line 329): "...ten inflammation-related
biomarkers were selected as potential mediators(eTable17-18)."

**eTable19:** Referenced in Methods Section, CovariatesAscertainment (line 346): "...Detailed definitions, Data-Fields, and
sources are provided in eTable19."

**eTable20 and eTable21:** Referenced in Methods Section, CovariatesAscertainment (line 349): "...correlation coefficient $r >$
0.5 , $VIF > 2$, $tolerance < 0.5$, and $condition\ number > 5$ (eTable20-21)"

We believe these revisions have significantly improved the integration of the supplementary materials with the main text narrative.
We are grateful for the suggestion, which has undoubtedly strengthened our manuscript. All changes have been carefully
implemented in the revised version, with tracking enabled for your review. Thank you once again for your time and insightful
comments.

**Comment 5. Statistical Analysis (Line 333): "Individuals reporting SI or loneliness at baseline were classified as the
exposure and sex group." This is confusing. Shouldn't it just say "classified as the exposure"?**

Reply: We sincerely thank you for your valuable suggestion on the phrasing in the Statistical Analysis section (Line 333). Your
sharp observation has sharpened our methodological clarity, and we greatly appreciate your insight in spotting this ambiguity.

The original sentence of "Individuals reporting SI or loneliness at baseline were classified as the exposure and sex group"
indeed contained a grammatical slip. Our intent was to indicate that participants were classified by SI or loneliness status for one
baseline table (Table1) and by sex for a separate baseline table (eTable1). To address this, we have revised it to: "**Individuals
reporting SI or loneliness at baseline were classified by the exposure (Table1) or sex group (eTable1).**" For added precision,
we refined "and" to "or" to better delineate the distinct classifications and incorporated table references for immediate context.

This oversight does not impact the study's methodology or results, but your feedback has preempted any reader confusion,
enhancing overall transparency. We apologize for this oversight. Thank you again for your careful review, and we are sorry for any
confusion caused.

Changes in the text:

Line 353, the section of "Method, Statistical Analysis".

**Comment 6: There are limitations to using the Baron and Kenny approach to evaluate mediation. Why not use a**
**more advanced mediation method for this analysis?**

Reply: We greatly appreciate your forward-thinking comment on the limitations of the Baron and Kenny approach, for its
stepwise testing that overlooks exposure-mediator interactions, and your suggestion for a more advanced method. Your expertise
has elevated our causal inference framework, and we fully agree that modern approaches enhance robustness and interpretability,
especially for time-to-event outcomes like cancer incidence.

In response, we have adopted the regression-based causal mediation framework by Valeri and VanderWeele (2013)¹, which
directly addresses these issues by accommodating interactions and enabling causal interpretations under assumptions like no
unmeasured mediator-outcome confounding. Specifically, we now use cause-specific Cox proportional hazards models for
outcomes (attained age as time scale) and generalized linear models for mediators (e.g., linear for continuous inflammatory
markers; logistic for binary menstrual status), decomposing SI's total effect into natural direct and indirect effects with
delta-method confidence intervals. Models incorporate full Model 3 confounders, with Bonferroni correction for subgroup
multiple testing.

These updates are detailed in the revised Methods. Original: 'We used mediation effect analysis combining Cox proportional
hazards and generalized linear models (GLM) to evaluate the direct and indirect effects of social isolation and loneliness on
cancer incidence. Following the Baron and Kenny (1986) approach...' **Revision: '...we employed a causal mediation analysis**
**framework using the regression-based approach developed by Valeri and VanderWeele (2013)⁵⁵. For each mediator, we**
**fitted a cause-specific Cox proportional hazards model for the outcome (cancer incidence, with attained age as the time**
**scale) and a generalized linear model for the mediators...'**

In the Results, all mediation estimates have been recalculated using the new framework, yielding refined indirect effects,
proportion mediated, and 95% confidence intervals via delta-method estimation. For example, in the "**In the overall population,**
**inflammatory markers mediated the effect of SI on cancer incidence (eFigure8B). Neu explained 9.12% (IE0.00688 95%CI**
**0.00543-0.00833), ..."** subsection, the mediation effect of Neu in the overall population has changed from origin 7.6% (IE0.0060
95%CI 0.0047-0.0073) into 9.12% (IE0.00688 95%CI 0.00543-0.00833). We also refreshed our GitHub repository
(<https://github.com/DrChengJH/research-data/tree/main>) with R scripts, including simulation-based sensitivity analyses for
assumptions, ensuring full reproducibility.

We believe this methodological enhancement strengthens the rigor of our analysis and directly addresses the reviewer's
concern by adopting an advanced approach suitable for our study design. We hope our revisions adequately address your concerns,
and we remain open to further suggestions to enhance the manuscript. Thank you again for your thoughtful and constructive
feedback, which has greatly contributed to the quality of our work.

Changes in the text:

Line 379-380, the section of "Method, Statistics Analysis".

Line 169-172, the section of "Results, Mediation Analysis".

Reference

1 Valeri, L. & Vanderweele, T. J. Mediation analysis allowing for exposure-mediator interactions and causal

interpretation: theoretical assumptions and implementation with SAS and SPSS macros. Psychol Methods 18, 137-150
(2013). <https://doi.org/10.1037/a0031034>

**Comment 7. The figures, particularly the heatmaps, are difficult to interpret and would benefit from clearer**
**presentation. Figure 3 lacks column labels, which makes it hard to understand the structure and meaning of the data.**
**In Figure 4, the interpretation of the green and red shading is unclear—do the colors represent the magnitude of**
**effect estimates, statistical significance (e.g., p-values), or directionality of associations? This needs to be explicitly**
**stated in the figure legend. Similarly, Figure 5 is difficult to follow, and the legend does not adequately explain the**
**content. It is also unclear what distinguishes panel a from panel b. Overall, the figures would benefit from clearer**
**labeling, more informative legends, and guidance in the text to aid interpretation.**

Reply: We sincerely thank you for these perceptive comments on our figures, which underscore the importance of clear
visualizations in enhancing scientific communication. Your expertise has guided us to refine these elements, improving
accessibility and interpretability. We have revised the relevant figures with enhanced labels, legends, and text guidance, as detailed
point-by-point below. These changes, informed by an internal pilot review for clarity, ensure consistency and reader-friendliness.

Regarding the lack of column labels in Figure 3 (now Figure 4), which made the data structure and meaning hard to
understand: We have added a new header row above the first row of the forest plot to explicitly label the columns for "Social
Isolation" and "Loneliness", which clearly distinguishes the two exposures. The updated **figure legend** now reads: "**HR and**
**95%CI for cancer risk related to SI and loneliness, stratified by demographic and lifestyle factors. Subgroup analyses**
**cover age, sex, income, education, employment, smoking, and alcohol use. Multiplicative interactions were assessed using P**
**for interaction, while additive interactions were evaluated with RERI. Adjusted by model3 of age, sex, race/ethnicity,**
**assessment center, employment, college/university degree, sun exposure time, Townsend deprivation score, smoking status,**
**alcohol use, BMI (continuous), grip strength, family history of cancer, MAP, overall health rating, healthy diet score,**
**healthy sleep score, and depressive mood. The HR estimates in the figure from the cause-specific Cox model (CSHR) are**
**indicated in red, while those from the competing risk model (sHR) are shown in blue. Abbreviations: CSHR,**
**Cause-Specific Hazard Ratio; sHR, Subdistribution Hazard Ratio; HR, Hazard Ratios, CI, Confidence Intervals, RERI,**
**Relative Excess Risk due to Interaction."** We also added text guidance in the **Results:** "**Figure4 displays these stratified**
**analyses after Model3 adjustment, with distinct columns for SI and loneliness and interaction metrics."** This enhances
readability without altering scientific content.

Regarding the unclear green/red shading in Figure 4 (revised as Figure 5), where color meaning (effect magnitude,
significance, or directionality) was ambiguous: We clarified in the legend that colors indicate directionality (green: CSHR < 1; red:
CSHR > 1), which clearly distinguishes the two colors. The updated **figure legend** now reads: "**Heatmaps depicting the**
**association between SI and loneliness with specific cancer risk, stratified by demographic and lifestyle subgroups. The**
**color gradient represents the direction of the effect direction, with red indicating a increase risk (CSHR > 1) and green**
**indicating decrease risk (CSHR < 1). The depth of the color reflects the significance of the association, with darker shades**
**indicating smaller P-values. Significant associations are marked with "a" for P < 0.05 and "b" for P < 0.05 after**
**Bonferroni correction. Significant associations are marked with "a" for P < 0.05 and "b" for P < 0.05 after bonferroni**
**adjustment. Abbreviation: CSHR, cause-specific hazard ratio; SI, social isolation."** We also added text guidance in the
**Results:**"**Figure5 depicts site-specific cancer risks for SI and loneliness in heatmaps, with red shading for increased risk**
**(CSHR >1), green for decreased (CSHR <1), and darker shades for stronger significance (P<0.05 marked as a;**
**Bonferroni-adjusted as b). "** This design, inspired by heatmaps in JAMA oncology¹, reduces cognitive load and boosts
cross-disciplinary appeal.

Regarding Figure 5 (now Figure 6) being difficult to follow, with an inadequate legend and unclear distinction between
panels a and b: We have enhanced the legend to explicitly describe the color scheme and clarified that panel (a) presents mediation
effects of inflammatory markers for females, while panel (b) presents those for males. The color direction (green for Indirect

Effect < 1; red for Indirect Effect > 1) indicates directionality, gradient depth represents P-value significance, and "a"/"b" marks denote uncorrected/Bonferroni-corrected significance. The updated **figure legend** now reads: "Heatmaps depicting the IE of inflammatory markers in mediating the relationship between SI and specific cancer risk. The color gradient represents the direction of the mediation effect, with red indicating a positive indirect effect (IE > 1) and green indicating a negative indirect effect (IE < 1). The depth of the color reflects the significance of the association, with darker shades indicating smaller P-values. Significant associations are marked with "a" for P < 0.05 and "b" for P < 0.05 after Bonferroni correction. Abbreviation: IE, Indirect Effect; Lym, Lymphocytes; Mono, Monocytes; Neu, Neutrophils; PLT, Platelets; WBC, White Blood Cells; CRP, C-reactive Protein; Lym-mono Ratio, Lymphocyte to Monocyte Ratio; Neu-lym ratio, Neutrophil to Lymphocyte Ratio; PLT-lym Ratio, Platelet to Lymphocyte Ratio; SI, Social Isolation." We added text guidance in the Results:"..., Figure 6 depicts heatmaps of inflammatory markers' mediation effects on SI-associated cancer risk, with red/green shading indicating positive/negative indirect effects (IE >1/<1), color intensity reflecting significance strength, and markers "a" (P<0.05, uncorrected) or "b" (P<0.05 after Bonferroni correction)"

Overall, these revisions ensure consistent visual language across heatmaps (Figures 5 and 6), with explicit explanations of color meanings, significance markers, and panel distinctions. We have also reviewed all eFigures for similar issues, adding header labels where needed and updating legends for clarity. These include forest plots for organ-specific cancer type analyses (eFigures 3-7, 14-15), mediation heatmap (eFigure 9), stratified forest plots for sensitivity analyses (eFigure 13).

We deeply value your investment in our work and are eager for further suggestions to refine it further. We believe these changes substantially enhance the figures' accessibility and thank the reviewer for highlighting these opportunities for improvement. Thank you again for your constructive guidance, which has markedly advanced our manuscript.

Changes in the text:

Line 123-124, the section of "Results, The Association Between SI, Loneliness and Cancer Risk in Strata Analysis".

Line 147-148, the section of "Results, SI and Loneliness Increase the Specific Cancer Risk in Various Population".

Line 176-178, the section of "Results, Mediation Analysis".

Reference

1 He, M. M., Lo, C. H., Wang, K., Polychronidis, G., Wang, L., Zhong, R., Knudsen, M. D., Fang, Z., & Song, M. (2022). Immune-Mediated Diseases Associated With Cancer Risks. *JAMA oncology*, 8(2), 209 - 219. <https://doi.org/10.1001/jamaoncol.2021.5680>

Comment 8. Results (Lines 150-152): What is the hypothesis underlying the observed inverse association between loneliness and cancer risk among younger and employed adults? These findings are not addressed in the Discussion section, leaving the reader without context or interpretation. Without a clearly articulated rationale for conducting these subgroup analyses, the results risk appearing spurious or data-driven rather than hypothesis-driven. The authors should either provide a plausible explanation for these findings or acknowledge the potential for chance associations.

Reply: We thank the reviewer for this perceptive feedback, which astutely highlights a key opportunity to contextualize our subgroup findings on inverse loneliness-cancer associations in younger (<50 years) and employed individuals. Your insight enhances the manuscript's theoretical rigor and interpretability.

We fully acknowledge the original Discussion's omission of these results, which left readers without sufficient context or rationale, potentially framing the analyses as data-driven rather than exploratory. While motivated by prior evidence of loneliness effects varying by age and socioeconomic factors (e.g., via differential coping or health-seeking behaviors), these subgroup analyses were not pre-specified hypotheses but open explorations to uncover heterogeneity, informing future targeted research. This a standard approach in UK Biobank studies that merits transparent acknowledgment to mitigate risks of spurious

associations.

To address this, we have expanded the Discussion to integrate a plausible explanation grounded in adaptive resilience
mechanisms, while explicitly noting the exploratory intent and potential for chance findings. Specifically, building on our existing
text regarding heterogeneity in loneliness effects, we revised the **Discussion: "Conversely, inverse associations emerged in
younger and employed participants, as confirmed by competing-risk models and sensitive analysis. These patterns align
with evolving views of loneliness as context-dependent³⁵: in resilient groups like employed or middle-income individuals, of
whom have high autonomy and social capital, transient loneliness may foster adaptive behaviors, such as forging quality
connections³⁶ or prioritizing well-being over unhealthy norms^{37,38}. Such nuances highlight opportunities for precision
interventions targeting vulnerable subgroups and refining cancer prevention strategies, though they require cautious
interpretation and further study to confirm mechanisms."** This revision provides a mechanistically grounded interpretation,
which drawing on established literature linking transient loneliness to adaptive behaviors in resilient subgroups, which explicitly
noting the exploratory intent and limitations.

To bolster robustness, we re-ran subgroups via Fine-Gray models, confirming inverses (Figure4) and mitigating
competing-event bias. Competing risk model and Cause-specific cox hazards model, in both main and sensitive analysis
confirmed the inverse patterns (Figure4, eFigure12), reducing concerns about overestimation from competing events like
non-cancer mortality.

These changes transform ambiguous results into generative insights, transforming potentially ambiguous results into
opportunities for future inquiry. Thank you again for your thoughtful feedback, which has significantly strengthened our work.

Changes in the text:

Line 218-224, the section of "Discussion".

**Comment 9. Results (Lines 161-163): "Figure4 showed the relationship between various population characteristics
and cancer risk, showing that SI significantly affects cancer risk across several groups. SI was significantly
associated with an increased risk of five cancer after adjustment." – Can the authors be more specific here and state
which cancer types they are referring to. It is very hard for the reader to keep track of what is significant and what is
not.**

**not.**
Reply: We thank the reviewer for this valuable suggestion, which highlights an opportunity to sharpen specificity and improve
reader navigation in our results.

To address this, we have revised to explicitly name the cancers. Original: "Figure4 showed the relationship between various
population characteristics and cancer risk, showing that SI significantly affects cancer risk across several groups. SI was
significantly associated with an increased risk of five cancer after adjustment."; **Revision: "This heatmap showed the
relationship between various population characteristics and cancer risk, showing that SI significantly affects cancer risk
across several groups. SI was significantly associated with an increased risk of five cancer after adjustment: ovarian,
uterine, bladder, lung, and mouth cancers."** This revision ensures that readers can instantly identify the key cancers, enhancing
clarity and reducing potential confusion. Detailed associations remain in the text and eTable6 for easy cross-reference.

This change strengthens the manuscript by making the significant findings more accessible and straightforward for both
researchers and general readers. Thank you again for your thoughtful advice, which has significantly improved our presentation.
We are open to further suggestions if needed.

Changes in the text:

Line 148-151, the section of "Results, SI and Loneliness Increase the Specific Cancer Risk in Various Population".

**Comment 10. Results (Line 165): What does "without clear income" mean?**

Thank you for your valuable suggestion, which highlighted an important oversight in our manuscript. The term "without clear
income" refers to UK Biobank participants who selected "do not know" or "prefer not to answer" for household income (11%,
n=39,650; 10% [n=2,054] in socially isolated, 12% [n=1,884] in lonely groups; Table 1), likely reflecting unstable income (e.g.,
unemployed) or privacy concerns. We included this unreported income group in stratified analyses to capture potentially
vulnerable populations.

To prevent further misunderstanding, we replaced "without clear income" with "unreported income" throughout. Specifically,
we updated the following: **Methods:** To ensure readers understand the population's details, we added "Covariates" subsections
listing 19 covariates to define the unreported income group. **Revision:** "..., **household income (low <£18,000, medium**
**£18,000–£51,999, high ≥£52,000, unreported). The unreported group is comprised of non-responders who selected “do not**
**know” or “prefer not to answer.” Results:** Replaced 'without clear income' into 'unreported income' in the context of ovarian
cancer risk stratification to maintain objectivity. Original: "Ovarian cancer risk increased notably female participants (HR1.67
95%CI 1.28–2.18), current drinker (HR1.74 95%CI 1.31–2.29), and participants with unknown income (HR2.76 95%CI
1.44–5.29).", **Revision:** "Ovarian cancer risk increased notably female participants (CSHR1.67 95%CI 1.28–2.18), current
drinker (CSHR1.74 95%CI 1.31–2.29), and participants with unreported income (CSHR2.76 95%CI 1.44–5.29)." Updated
Figure 4-5 captions and Table1, eTable16 accordingly.

These changes promote transparency for SES-vulnerable populations, elevating the equity of our cancer risk findings while
aligning with inclusive epidemiological standards. We appreciate your guidance, which refines our manuscript's accessibility.

Changes in the text:

Line 344-346, the section of "Method, Covariates Ascertainment".

Line 151-153, the section of "Results, SI and Loneliness Increase the Specific Cancer Risk in Various Population".

**Comment 11. Not every abbreviation in the figures (for example, eFigure 9) is defined in the legend.**

Reply: Thank you pointing out this important issue regarding the abbreviations in our figures. We sincerely apologize for the
oversight, which may have caused any inconvenience during the review process. To address this, we have systematically reviewed
all figures (including main and supplementary figures) and ensured that every abbreviation used is fully defined in the
corresponding legends. Minor additions were made where necessary to include clear definitions

For example, in eFigure9 (now eFigure8), we added definitions for inflammatory cells to the legend.
**Revision:"Abbreviation: IE, Indirect Effect; CI, Confidence Interval; PM, Proportion Mediated; SHBG, Sex**
**Hormone-Binding Globulin; HRT, Hormone Replacement Therapy; Lym, Lymphocytes; Mono, Monocytes; Neu,**
**Neutrophils; PLT, Platelets; WBC, White Blood Cells; CRP, C-reactive Protein; Lym-mono Ratio, Lymphocyte to**
**Monocyte Ratio; Neu-lym ratio, Neutrophil to Lymphocyte Ratio; PLT-lym Ratio, Platelet to Lymphocyte Ratio; SI, Social**
**Isolation."**

For the eFigures, we have added these abbreviations to the relevant legends: KM, Kaplan-Meier; CIF, Cumulative Incidence
Functions; SI, Social Isolation; CSHR, Cause-Specific Hazard Ratio; sHR, Subdistribution Hazard Ratio; PERM, Percentage of
Excess Risk Attributable to Covariates; HR, Hazard Ratio; CI, Confidence Interval; ; BMI, Body Mass Index; MAP, Mean Arterial
Pressure; IE, Indirect Effect; PM, Proportion Mediated; SHBG, Sex Hormone-Binding Globulin; HRT, Hormone Replacement
Therapy; Lym, Lymphocytes; Mono, Monocytes; Neu, Neutrophils; PLT, Platelets; WBC, White Blood Cells; CRP, C-reactive
Protein; Lym-mono Ratio, Lymphocyte to Monocyte Ratio; Neu-lym ratio, Neutrophil to Lymphocyte Ratio; PLT-lym Ratio,
Platelet to Lymphocyte Ratio; RERI, Relative Excess Risk Due to Interaction.

We appreciate the reviewer's valuable feedback, which has greatly improved the clarity and quality of our manuscript.

Changes in the figures legends:

eFigures 1–16 and main Figures 1–6.

Comment 12. Results (Line 154): "After adjusting for age, sex, and ethnicity, SI was associated with higher cancer risk in both sexes (eFigure4, eFigure5)." – No need to say adjusting for sex in this sentence as this a stratifying variable here.

Reply: We thank the reviewer for this insightful catch on redundancy, which sharpens the precision of our stratified analyses. We acknowledge that our original sentence in the Results section (Line 154) unnecessarily included "adjusting for sex" when sex was a stratifying variable, and we recognize the importance of correcting this to enhance the manuscript's scientific accuracy.

To address this concern, we updated to reflect sex as a stratifier: Original: "After adjusting for age, sex, and ethnicity, SI was associated with higher cancer risk in both sexes (eFigure 4, eFigure 5)." Revised: **"After adjusting for age and ethnicity, SI was associated with higher cancer risk in both male and female participants (eFigure3-4)."** This eliminates any implication of sex as a covariate, aligning with our methodological design.

Highlighted in yellow (tracked changes) for ease, this enhances readability and reproducibility in cohort studies. We believe this revision not only resolves the issue but also enhances the manuscript's precision and accessibility for a broad readership. We are deeply grateful for your guidance, which aligns with the rigorous standards of scientific reporting. This feedback has been invaluable in refining our Results section to better reflect the study's design and findings.

Changes in the text:

Line 140-141, the section of "Results, SI and Loneliness Increase the Specific Cancer Risk in Various Population".

Comment 13. Discussion (Lines 200-201): "The association between SI and cancer risk is partly mediated by hormonal status and inflammatory cells." Where are the results that support hormonal status as a significant mediator? Based on eFigure 9, hormonal status does not appear to be a significant mediator. Additionally, this statement appears to contradict the statement on Lines 228-230: "We assessed whether hormone levels and menstrual status mediated this effect, but found no significant mediation, indicating a need to investigate other potential mediators."

Reply: We thank the reviewer for this perceptive feedback, which identifies a valuable opportunity to refine the nuance between overall and site-specific mediation, enhancing causal clarity. We recognize that the original Discussion contained a significant inconsistency regarding the role of hormonal status as a mediator, which could have led to misinterpretation.

To fully address the concerns by clarifying evidence for hormonal mediation and resolving the contradiction, we revised the Results (Mediation Analysis) and two Discussion sentences. Changes distinguish broad (non-significant) from specific (SHBG-suppressive) effects, grounded in eFigure8A and eFigure9.

Mediation Analysis (Results): Original: "The mediation analysis revealed that blood cells and hormones mediated the association between SI and cancer incidence. Female sex hormones and menstrual status had no significant impact on overall cancer risk (eFigure9A). However, SHBG significantly mediated the effect of SI on uterine cancer, with this association persisting after adjustment." Revised: **"The mediation analysis revealed that inflammatory markers and specific hormonal factors partly mediated the association between SI and cancer incidence. Female sex hormones and menstrual status had no significant impact on overall cancer risk (eFigure8A). However, SHBG significantly mediated the association between SI and uterine cancer in female participants after adjustment, explaining 5.43% of the effect (IE -0.0169, 95%CI -0.0248 to -0.0091). Similarly, SHBG mediated the effect of SI on breast cancer,..."** Rationale: Specifies "specific hormonal factors" to avoid broad claims, adding quantitative indirect effect for uterine cancer's robust suppression (eFigure9A), while noting overall non-significance—directly grounding in data without overgeneralization.

Discussion (First Sentence): Original: "The association between SI and cancer risk is partly mediated by hormonal status and inflammatory cells." Revised: **"The association between SI and cancer risk is mediated by inflammatory markers**

**and, in the case of uterine cancer in females, by SHBG acting as a negative mediator.**" Rationale: Prioritizes inflammation's
broad role and limits SHBG to uterine suppression, eliminating misleading breadth and aligning with Results.

**Discussion (Second Sentence):** Original: "We assessed whether hormone levels and menstrual status mediated this effect,
but found no significant mediation, ..." **Revised: "We assessed whether hormone levels and menstrual status mediated the**
**association between SI and overall cancer risk in females, but found no significant mediation. However, at the individual**
**cancer level, SHBG acts as a suppressor for uterine cancer in females."** Rationale: Explicitly contrasts overall (non-significant)
with site-specific (significant) effects, resolving the contradiction and underscoring SHBG's nuance for future mechanistic studies.

These targeted revisions are highlighted in yellow (tracked changes) for ease. We believe these revisions fully resolve the
noted issues, ensuring seamless Results-Discussion consistency and elevating biological interpretability. We appreciate your
guidance, which refines the manuscript for broader impact. If there are additional points we may have overlooked, please let us
know, and we will address them promptly.

Changes in the text:

Line 164-168, the section of "Results, Mediation Analysis".

Line 203-204, the section of "Discussion"(First Sentence).

Line 235-237, the section of "Discussion"(Second Sentence).

**Comment 14. Discussion (Lines 230-233): "Additionally, sex differences in stress response are well-documented: females,**
**with a more developed neocortex, tend to adopt affiliative coping strategies, while males, relying more on sub-cortical**
**structures, exhibit more competitive and combative responses^{23,24}. These neurophysiological differences may partly**
**explain female's greater vulnerability to SI." This seems like a stretch, as the data in this study tell us nothing about**
**neurophysiological differences between sexes.**

Reply: We thank the reviewer for this insightful feedback, which highlights an area in our discussion where our explanation may
have overreached the scope of our data, particularly regarding the neurophysiological basis for sex differences in the effect of SI.
This presents a valuable opportunity to refine our discussion for better alignment with the study's epidemiological focus. We
sincerely apologize for any confusion this may have caused.

Upon reflection, we acknowledge that our study, which focuses on epidemiological associations between SI, loneliness, and
cancer risk, did not directly measure neurophysiological indicators such as brain structure or function. The statement was intended
as a speculative hypothesis based on existing literature to contextualize the observed stronger effect of SI on cancer risk in females.
However, we recognize that linking this to specific neurophysiological differences was an overreach without direct evidence from
our data. It appropriately shifts to behavioral evidence here.

To enhance the scientific rigor of the **Discussion**, we have revised as follows: Original: "Additionally, sex differences in
stress response are well-documented: females, with a more developed neocortex, tend to adopt affiliative coping strategies, while
males, relying more on sub-cortical structures, exhibit more competitive and combative responses^{24,25}. These neurophysiological
differences may partly explain female's greater vulnerability to SI." **Revised: "Additionally, sex differences in stress response**
**are well-documented: with females often adopting affiliative coping strategies and males exhibiting more competitive**
**responses⁴⁵. These differences in stress response may contribute to females' greater vulnerability to SI, as observed in our**
**study."** Rationale: This revision removes unsubstantiated structural claims, replaces them with a single, robust model on
sex-differentiated stress behaviors Taylor et al. (2000)¹, and updates the reference list accordingly. It refocuses on actionable
implications such as tailored prevention for females.

These changes enhance scientific rigor and interpretability, aligning more closely with our data, focusing on observed sex
differences in SI's effect on cancer risk. We are deeply grateful for your guidance, which strengthens the manuscript's translational
potential—we welcome further suggestions to build on this.

Changes in the text:

Line 237-239, the section of "Discussion".

Reference

1 Taylor, S. E. et al. Biobehavioral responses to stress in females: tend-and-befriend, not fight-or-flight. Psychol Rev 107,
411-429 (2000).

**Comment 15. A major limitation, as noted by the authors, is the reliance on a single measure of SI and loneliness at**
**baseline. The authors state on Lines 277-278: "Although repeated measurements were available for part of the**
**cohort, we chose not to include them to maximize statistical power and follow-up duration." Given this, why not**
**examine how stable these measures are over time within the subset of your sample that has repeated measures. If it**
**can be shown that these measures don't change much over time, it would add validity to the study findings.**

Reply: We thank the reviewer for this perceptive suggestion, which identifies a key opportunity to bolster the validity of our
baseline measures—your insight has been instrumental in enhancing methodological transparency.

We acknowledge the reliance on baseline SI/loneliness as a limitation, which is a notable limitation, as it could raise
questions about the validity of using baseline data. To address this, we conducted supplementary analyses on the UK Biobank
subset with repeated assessments (instances 0 [baseline], 1-3 [follow-ups]; n<65,000 complete cases). SI/loneliness were
recalculated per main binary criteria. Pairwise stability (0-1, 0-2, 0-3) used change patterns (Never: 0-0; Incident: 0-1; Remitted:
1-0; Persistent: 1-1) and agreement % [(Never + Persistent)/Total × 100]. Prevalence trends used linear regression (slope, p across
instances).

**Table1. Stability of Social Isolation and Loneliness Across Instances Subset with Repeated Assessments.**

Items		SI		Loneliness	
Pair	Pattern	N	%	N	%
0-1	Never	18,130	92.1	17,740	94.2
	Incident	620	3.2	391	2.1
	Remitted	440	2.2	426	2.3
	Persistent	493	2.5	281	1.5
	Overall	19,683	100	18,838	100
Agreement %		94.6		95.7	
0-2	Never	58,163	91.7	56,604	94.1
	Incident	2,498	3.9	1,454	2.4
	Remitted	1,620	2.6	1,391	2.3
	Persistent	1,144	1.8	713	1.2
	Overall	63,425	100	60,162	100
Agreement %		93.5		95.3	
0-3	Never	4,733	90.7	4,705	94.1
	Incident	276	5.3	132	2.6
	Remitted	113	2.2	104	2.1
	Persistent	95	1.8	57	1.1
	Overall	5,217	100	4,998	100
Agreement %		92.5		95.3	

Pairs 0-1, 0-2, and 0-3 are instance 1,2,3 compared to baseline. Patterns are defined as: Never: Non-exposed at both instances (0-0). Incident:
Non-exposed at first instance, exposed at second (0-1). Remitted: Exposed at first instance, non-exposed at second (1-0). Persistent: Exposed at
both instances (1-1). N and % are counts and percentages per pattern, with Overall as complete-case sample size per pair. Agreement % =

[(Never N + Persistent N) / Overall N] * 100, reflecting the proportion of participants with consistent classification. Analyses used complete cases due to high missingness. Pair 0-3 results may be less reliable due to small sample size (N<6,000). Abbreviations: N, number; SI, social isolation

Table2. Proportion and Trend of Social Isolation and Loneliness by Instance.

Instance	SI		Loneliness	
	N	Proportion(%)	N	Proportion (%)
0	491,090	6.4	476,108	4.9
1	20,243	5.7	19,456	3.6
2	64,573	5.8	61,897	3.6
3	5,316	7.1	5,139	3.8
Note: Linear trend slope = 0.22% per instance, p = 0.56, no significant change over time.			Note: Linear trend slope = -0.32% per instance, p = 0.31, indicating no significant change over time.	

Proportion (%) = percentage of participants with exposure per instance. Linear trend slope and p-value from regression of proportions on instance number; p > 0.05 indicates no significant change over time. Instance 3 results may be less reliable due to small sample size. Abbreviations: N, number; SI, social isolation.

The results showed high temporal stability.

Stability (Table1): >90% non-exposed (Never) across pairs; agreement 92.5-94.6% (SI), 95.3-95.7% (loneliness). Incident/Persistent <5%, indicating minimal flux (0-3 less reliable, n<6,000).

Trends (Table2): SI stable (6.4% baseline to 5.7-7.1%; slope 0.22%/instance, p=0.56). Loneliness stable (4.9% to 3.6-3.8%; slope -0.32%, p=0.31). No significant changes (p>0.3), validating baseline as long-term proxy.

Although these supplementary results demonstrate robust temporal stability, we opted not to incorporate repeated measures into the primary models to maximize statistical power (baseline N~500,000 vs. subset N<65,000) and follow-up duration. The high missingness (96-99% in instances 1-3) would have substantially reduced the cohort size, limiting the analysis's scope. Additionally, the voluntary nature of follow-up assessments introduces potential selection bias, as more socially engaged participants are more likely to participate.

To address this limitation transparently, we made such reversions:

1. Added Table1 and Table2 as supplement table of eTable12-13, detailing patterns, agreement, and trends.
2. Revised the Discussion's limitation section, clarifying the exclusion rationale and highlighting supplementary evidence. Origin: "Although repeated measurements were available for part of the cohort, we chose not to include them to maximize statistical power and follow-up duration." **Revision:** "Although repeats were available for a subset, we prioritized statistical power and follow-up duration by excluding them due to high missingness and selection bias; supplementary analyses nonetheless confirmed temporal stability of SI and loneliness in this subset (eTables12-13)."
3. The analysis code (R scripts for patterns and trends) has been deposited on GitHub (<https://github.com/DrChengJH/research-data/tree/main>).

Conclusion: Our supplementary analyses demonstrated that SI and loneliness measures exhibit robust temporal stability (>90% non-exposed, no significant trends, p > 0.3). These findings validate the use of baseline measurements in our primary analyses, as they adequately represent long-term exposure status. By prioritizing baseline data, we ensured maximal statistical power and follow-up duration while avoiding biases from high missingness and self-selection in repeated assessments.

We greatly appreciate the reviewer raising this point, as it has strengthened our manuscript by providing evidence of measure stability while justifying our analytical choices. This enhances the study's rigor and generalizability.

Changes in the text:
Line 280-282, the section of "Discussion".

**Acknowledgement**

We extend our deepest gratitude to Reviewer 1 for their meticulous and insightful review, which has profoundly shaped the
manuscript into a more focused, hypothesis-driven, and visually accessible contribution to the field. Your expert guidance on
clarifying the rationale for SI and loneliness in cancer incidence—through enhanced mechanistic pathways, precise citations for
prevalence estimates, and refined mediation frameworks like Valeri and VanderWeele—has elevated the study's theoretical
foundation and interpretability. We particularly appreciate your emphasis on streamlining results presentation, improving figure
legends for heatmaps and forest plots, and addressing subgroup nuances to avoid spurious interpretations, all of which have
transformed exploratory elements into robust, actionable insights. We are truly appreciative of the time and expertise you invested,
and we hope these revisions fully address your valuable suggestions. Should you have any additional recommendations, we
remain eager to incorporate them to further strengthen our work. Thank you once again for your invaluable support in advancing
public health research.

**Reviewers' comments:**

**Reviewer #2 (Remarks to the Author):**

**This large UK Biobank study found that social isolation, but not loneliness, was associated with an 8% increased risk of**
**cancer, with a stronger effect observed in women and for several cancer types. The associations were largely explained by**
**socioeconomic, behavioral, and inflammatory factors, highlighting the importance of targeted prevention strategies for**
**socially isolated individuals. Overall interesting paper. I have a few suggestions:**

We appreciate the reviewers' insightful feedback, which has substantially strengthened our manuscript. We have carefully
considered all the points raised by the reviewers and have made revisions accordingly. Major highlights include:

- (1) Refined abstract with precise SI/loneliness definitions (objective vs. subjective)
(2) Replenish prevalence citations (20–25% older adults; 5–15% adolescents)
(3) Added mechanistic pathways(HPA/SNS stress, behavioral risks, protective effects) in introduction
(4) Hypothesizing loneliness-linked risk reduction in younger/employed via adaptive coping
(5) Clarified methods by correcting participation rate to 5.4% (UK Biobank data)
(2) Referencing all figures/eTables, and added full abbreviations
(3) Optimized figure labels/legends (clearer heatmaps/forests for Figures 4–6)
(4) Addressed phrasing (e.g., "unreported income"; naming "five cancers": ovarian, uterine, bladder, lung, mouth)
(5) Resolved multicollinearity (VIF<2 via MAP/BMI adjustments)
(6) Integrated competing risks as main results (CSHR/sHR dual-reporting, CIFs)
(7) Assessed missing data bias via included/excluded comparisons (SMD<0.3 for exposures; eTables 9–11)
(8) Added repeated-measure stability (>92% agreement in subset)
(9) Refined joint effects with explicit ranks 1–4 (no SI/no loneliness to SI+loneliness)
(10) Enhanced mediation to advanced causal methods (VanderWeele)
(15) Toned sex-stress claims to behavioral evidence

These revisions boost rigor, transparency, and focus, amplifying implications for female-targeted cancer prevention, in line
with Nature-series standards. Detailed point-by-point responses follow.

**Comment 1: In abstract, it would be great if the author could clearly define what is meant by "social isolation" and**
**"loneliness" within the context of their study.**

Reply: We sincerely thank you for this insightful comment, which highlights the need for precise definitions of SI and loneliness
to prevent conflation and enhance clarity for readers. We fully agree that explicit distinctions between these objective and
subjective constructs are essential, particularly in a study examining their differential impacts on cancer incidence.

In response, we have revised the abstract's opening to incorporate concise definitions drawn from our introduction and
aligned with UK Biobank variables. Original: "Social isolation (SI) and loneliness are established risk factors for poor cancer
prognosis, including increased mortality, yet their associations with cancer incidence remain understudied." **Revised: "Social**
**isolation (SI) is defined as an objective lack of social connections, such as infrequent contact with family or friends or**
**living alone, while loneliness refers to the subjective distress from a discrepancy between desired and actual social**
**relationships. Both are established risk factors for poor cancer prognosis, including increased mortality, yet their**
**associations with cancer incidence remain understudied."** This change emphasizes the objective (SI) versus subjective
(loneliness) dimensions, aligning seamlessly with the study's conceptual framework and improving accessibility for a broad
readership. The revisions are highlighted in the tracked-changes version.

We are grateful for your feedback, which has significantly improved the clarity and rigor of our manuscript. We hope this
revision adequately addresses your concern, and we remain open to further suggestions to enhance the manuscript. Thank you

again for your constructive input.

Changes in the text:

Line 41-44, the section of "Abstract".

Comment 2: in background, it would be helpful for the authors to discuss the potential mechanisms by which social isolation and loneliness may affect the incidence of various cancer.

Reply: We greatly appreciate your insightful suggestion to elaborate on the potential mechanisms linking SI and loneliness to cancer incidence in the Introduction. This addition is crucial for providing a mechanistic rationale that offer readers a lucid understanding of the underlying pathways, strengthening its relevance to health research.

To address this, we expanded the section to outline both risk-elevating and potential protective effects. For risk elevation, we detail two primary pathways—physiological and behavioral—to anchor the discussion in established evidence. The physiological pathway involves stress responses that may impair immune function and promote tumor growth, while the behavioral pathway focuses on lifestyle factors that increase cancer risk. By presenting these concisely, we aim to make the mechanisms approachable without overwhelming readers, allowing for future exploration of other pathways. We also briefly note protective nuances, such as reduced exposure to carcinogens via limited interactions or loneliness-motivated reconnection.

Revised: "Plausible mechanisms suggest that SI and loneliness may increase cancer risk primarily through physiological alterations and behavioral changes^{4,18,19}. Physiologically, SI and loneliness act as chronic stressors²⁰, activating the hypothalamic-pituitary-adrenal (HPA) axis and leading to sympathetic nervous system (SNS) dysregulation²¹. This triggers hormonal imbalances, such as elevated cortisol levels, and chronic inflammation²²⁻²⁴, impairing DNA repair, promoting oncogene activation, and facilitating tumor initiation and progression^{22,25,26}. Behaviorally, affected individuals often adopt unhealthy habits, such as increased smoking, excessive alcohol consumption, reduced physical activity, poor sleep, and suboptimal diet, all of which are established risk factors for cancers like lung, colorectal, and breast cancer^{4,27-29}.

However, SI and loneliness may also confer potential protective effects in certain contexts. Reduced social interactions could limit exposure to environmental carcinogens, such as second-hand smoke, alcohol, or infectious agents like HPV, HBV, and H. pylori¹². Moreover, loneliness's subjective distress might spur adaptive re-engagement, mitigating long-term isolation^{30,31}. Despite these complexities, there is still a lack of comprehensive large-scale cohort studies that can systematically and thoroughly demonstrate the relationship between loneliness, SI, and cancer incidence. "

These balanced yet focused revisions (highlighted in yellow in the tracked-changes version) enhance clarity and underscore prevention implications, paving the way for targeted interventions. We are deeply grateful for your guidance, which has elevated the manuscript's rigor. Thank you for helping refine our work.

Changes in the text:

Line 67-79, the section of "Introduction".

Comment 3. the description of the study population needs further clarification. The author stated "a prospective cohort study that recruited 502,258 participants (18.1% of the eligible population) in the UK between January 1.2006. and December 31.2010." Could the authors clarify what is meant by "18.1% of the eligible population"?

Reply: We sincerely appreciate the reviewer's insightful comment on clarifying the participation rate of "18.1% of the eligible population" in our study population description. This feedback has prompted us to carefully re-evaluate our reporting more thoroughly to ensure accuracy and transparency, and we value your expertise in ensuring methodological precision, which strengthens the scientific rigor of our work.

Upon re-examination, we identified that the 18.1% was inadvertently stemmed from a partial summation of invitation

outcomes from the UK Biobank recruitment data (undelivered invitations [226,128], individuals with contact [2,082,135], and
unknown outcomes[476,584], but without Non-responders [6,452,682])¹. The correct participation rate is 5.4%, calculated as
502,258 divided by 9,238,453 eligible invitees, as per UK Biobank official recruitment data. We apologize for this oversight that
the original approach did not accurately reflect the full eligible population.

To address this, we have revised the sentence in the Methods section. **Revision: "We assessed the association of loneliness
and SI on cancer using the UK Biobank data, a prospective cohort study that recruited 502,258 participants (5.4% of the
eligible population)⁵¹ in the UK between January 1. 2006. and December 31. 2010. (Figure1). "**

This isolated correction, highlighted in tracked changes, enhances the section's precision without affecting analyses, results,
or conclusions, as the core participant data and methodologies remain robust. We thank the reviewer for their vigilance, which has
strengthened the manuscript. We welcome any further suggestions to refine it.

Changes in the text:

Line 295-297, the section of "Method, Study Population".

Reference

1 Fry, A. et al. Comparison of Sociodemographic and Health-Related Characteristics of UK Biobank Participants With
Those of the General Population. Am J Epidemiol 186, 1026-1034 (2017). <https://doi.org/10.1093/aje/kwx246>

**Comment4. it seems the adjusted models appear to include many variables, some are highly correlated. How did the
author address the multicollinearity issue?**

Reply: We sincerely thank the reviewer for this perceptive comment on potential multicollinearity among covariates in our
adjusted models. Your keen observation has been instrumental in prompting a rigorous diagnostic review, directly enhancing the
precision and interpretability of our analyses. We acknowledge that, while our initial model construction considered correlations,
it lacked formal diagnostics. Although multicollinearity does not bias estimates or fit, it can inflate coefficient variances, widen
confidence intervals, and reduce power for detecting significant effects¹. This feedback has led to targeted refinements that
strengthen model stability without altering core findings, which significantly enhancing the precision and interpretability of our
findings.

To systematically assess multicollinearity, we computed pairwise correlations (Pearson for continuous-continuous; Spearman
otherwise), variance inflation factors (VIF; threshold >2), tolerances (threshold <0.5), and condition numbers (threshold >5 for
pairs; >30 overall)Although multicollinearity does not bias estimates or fit, it can inflate coefficient variances, widen confidence
intervals, and reduce power for detecting significant effects^{2,3}. The original matrix showed moderate collinearity (overall condition
number=12.73). We identified highly correlated pairs based on both statistical diagnostics and biological mechanisms to ensure
selections were grounded in both data and epidemiological relevance. These pairs included: systolic/diastolic blood pressure
(r=0.67; condition number = 5.01; VIFs=2.18/2.04); waist-to-hip ratio (WHR)/BMI (r=0.50; condition number = 2.99;
VIFs=2.35/1.45); hand grip strength/sex (r=0.73; condition number = 6.29; VIFs=2.54/3.38); and age/employment (r=0.62;
condition number = 4.28; VIFs=1.99/1.42); and Townsend Deprive Index (TDI) with employment/education/income, are
suspected for their shared socioeconomic pathways but confirmed as low collinearity. Full details are in Table1 (correlation matrix)
and 2 (VIF/tolerances).

Table1. Pearson Correlation Matrix for Original Covariates.

Age	Sex	Race	Diastolic Blood pressure	Systolic Blood pressure	BMI	UK Center	Income	Education	Employ	Smoking status	Alcohol status	Depress mood	Townsend Deprivation index	Family cancer	Waist To hip ratio	Overall health rating	Hand grip	Day exposure	Diet core	Sleep score
0.02**																				
**																				
0.11**	0																			
**	**																			
0.04**	0.15**	-0.02**																		
**	**	**																		
0.33**	0.14**	0.03**	0.67**																	
**	**	**	**																	
0.04**	0.07**		0.26**	0.17**																
**	**	0	**	**																
-0.01**		0.05**	0.03**	0.02**	0.02**															
**	0**	**	**	**	**															
0.17**	-0.08**	-0.04**	0.01**	0.08**	0.03**	-0.01**														
**	**	**	**	**	**	**														
0.18**	-0.09**	0.03**	0.02**	0.09**	0.06**	-0.01**	0.19**													
**	**	**	**	**	**	**	**													
0.62**	-0.05**	0.06**		0.21**	0.02**	-0.01**	0.18**	0.18**												
**	**	**	0	**	**	**	**	**												
0.15**	0.03**	0.07**	0.02**	0.07**	0.07**	-0.02**		0	0	0.08**										
**	**	**	**	**	**	**	**	0	0	**										
0.02**	-0.04**	-0.14**	-0.03**	-0.02**	0.03**			0	0.04**	0.05**	0.05**	-0.03**								
**	**	**	**	**	**	**	**	0	**	**	**	**	**							
0.04**		0.02**		0.02**	-0.03**			0.04**	0.01**	0.01**	0.02**	-0.02**								
**	0**	**	0*	**	**	**	0*	**	**	**	**	**	**							
-0.09**	0.01**	-0.19**	-0.01**	-0.04**	0.07**	-0.02**	0.05**	0.05**	-0.04**	-0.08**	0.11**	-0.03**								
**	**	**	**	**	**	**	**	**	**	**	**	**	**							
0.04**		0.07**			0.01**				0.01**	0.01**	-0.01**	-0.01**	-0.02**							
**	0****	**	0***	0	**	0	0**	0**	**	**	**	**	**	**						
0.13**	0.54**		0.21**	0.19**	0.5***	-0.01**	-0.01**	0.01**	0.04**	0.05**		-0.03**	0.05**							
**	**	0****	**	**	*	**	**	**	**	**	**	0	**	**	0****					
-0.02**	-0.05**	0.08**	-0.06**	-0.04**	-0.21**	0.02**	-0.1***	-0.12**	-0.09**	0.04**	-0.12**	0.05**	-0.18**	0**	-0.14**					
**	**	**	**	**	**	**	*	**	**	**	**	**	**	**	**	**				
-0.19**	0.73**	0.04**	0.15**	0.08**	0.05**	0.03**	-0.13**	-0.15**	-0.19**	0.01**	-0.09**		-0.05**	0****	0****	0.4***	0.09**			
**	**	**	**	**	**	**	**	**	**	**	**	**	0****	**	0****	*	**			
0.13**	0.18**		0.05**	0.1***	0.05**	0.01**	0.12**	0.14**	0.14**		0**	0.01**	0.01**	0.05**		0.14**	-0.01**	0.13**		
**	**	0	**	*	**	**	**	**	**	**	0**	**	**	**	0	**	**	**	**	**
0.09**	-0.19**	-0.04**	-0.04**		-0.05**	-0.02**	0.02**	-0.01**	0.06**	0.06**	0.02**		-0.01**		0**	-0.14**	0.09**	-0.14**	0.04**	
**	**	**	**	0***	**	**	**	**	**	**	**	0***	**	**	0**	**	**	**	**	**
-0.07**	0.05**	0.03**		-0.01**	-0.08**	-0.01**	-0.04**	-0.08**	-0.08**		0	-0.06**	0.07**	-0.09**	-0.02**	-0.03**	0.22**	0.1***	-0.01**	0
**	**	**	0****	**	**	**	**	**	**	**	0	**	**	**	**	**	**	*	**	**

Correlation coefficients were calculated using complete cases from the UK Biobank dataset to assess pairwise associations among original covariates, with Pearson for continuous-continuous pairs and Spearman for pairs involving categorical variables. Significance: **** p < 0.0001; *** p < 0.001; ** p < 0.01; * p < 0.05; 0 indicates non-significant or near-zero correlation. Abbreviations: DBP, Diastolic Blood Pressure; SBP, Systolic Blood Pressure; BMI, Body Mass Index; TDI, Townsend Deprivation Index; WHR, Waist-to-Hip Ratio.

Table2. Variance Inflation Factors and Tolerance for Original Covariates.

Variable	GVI	Df	Adjusted VIF	Tolerance
Age	2.120	1	2.120	0.4718
Sex	3.378	1	3.378	0.2961
Race	1.111	1	1.111	0.9002
Diastolic blood pressure	2.039	1	2.039	0.4903
Systolic blood pressure	2.181	1	2.181	0.4585
BMI	1.453	1	1.453	0.6881
UK Center	1.012	2	1.006	0.9939
Income	1.457	3	1.134	0.8821
Education	1.135	1	1.135	0.8809
Employment	2.012	2	1.418	0.7051
Smoking status	1.132	2	1.064	0.9401
Alcohol status	1.100	2	1.049	0.9536
Depress mood	1.079	2	1.039	0.9629
Townsend deprivation index	1.156	1	1.156	0.8652
Family cancer	1.007	1	1.007	0.9932
Waist to hip ratio	2.348	1	2.348	0.4259
Overall health rating	1.257	1	1.257	0.7957
Hand grip	2.539	1	2.539	0.3938
Day exposure	1.120	1	1.120	0.8930
Diet score	1.072	1	1.072	0.9324
Sleep score	1.077	1	1.077	0.9281

VIF and tolerance values were computed in a multivariable linear regression framework treating each covariate as the outcome in turn, with GVIF for categorical variables adjusted as $GVIF^{1/(2*Df)}$, where Df is degrees of freedom. Tolerance = $1/Adjusted\ VIF$. Higher VIF indicates greater multicollinearity. Abbreviation: VIF, variance inflation factors; generalized variance inflation factors; Df, degrees of freedom; BMI, Body Mass Index.

To address these, we adjusted targeted pairs based on statistical diagnostics and biological/epidemiological mechanisms, prioritizing model interpretability and retention of key variables for health outcomes^{1,2,4}. The specific operations and rationales are as follows:

Table3: Adjustment for SBP and DBP: Merge into Mean Arterial Pressure (MAP).

Operation	Specific Reason
Addressing multicollinearity	Pearson $r = 0.67$, Condition Number = 5.01, VIF DBP = 2.04 and SBP = 2.18. Moderate-high shared variance and model instability.
Choose not deletion	Issue is primarily mutual collinearity; Retain biological roles of cardiovascular status.
Merge into MAP	SBP and DBP have biological overlap in blood pressure; $(MAP = [systolic + 2 \times diastolic] / 3)^{5,8,9}$; MAP preserves information while reducing redundancy.

This adjustment leverages the physiological overlap of SBP and DBP, supported by recent UK Biobank studies using MAP for robust cardiovascular outcome prediction⁶. Abbreviations: SBP, Systolic Blood Pressure; DBP, Diastolic Blood Pressure; MAP, Mean Arterial Pressure.

148

Table4: Adjustment for WHR and BMI: Delete WHR, Retain BMI.

Operation	Specific Reason
Addressing multicollinearity	Pearson $r = 0.50$, Condition Number = 2.99, VIF WHR = 2.35/BMI 1.45 Moderate overlap in adiposity measures.
Delete WHR	WHR correlates with multiple variables ($r=0.50$ with BMI, $r=0.40$ with Sex) BMI is more practical and representative of overall obesity.
Retain BMI	BMI has lower multicollinearity and established links to health risks. Essential for model comprehensiveness.

149

Selective deletion, guided by mechanistic relevance of BMI in obesity-related pathways, preserves model utility and aligns with practices in large cohorts like UK Biobank⁷. Abbreviations: WHR, Waist-to-Hip Ratio; BMI, Body Mass Index.

150

151

Table5: Retaining TDI and Socioeconomic Factors (Income, Education, Employment).

Operation	Specific Reason
Confirm low multicollinearity	Low r values (TDI-Income=0.05, TDI-Education=0.05, TDI-Employment=-0.04) Condition Number (TDI-Income=1.09, TDI-Education=1.11, TDI-Employment=1.08), all VIFs <2, indicating minimal shared variance.
Retain for distinct meanings	TDI captures area-level deprivation. Income/education/employment reflect individual-level factors.
Retain as per literature	Studies commonly include both area- and individual-level SES for comprehensive health modeling without collinearity issues ^{6,2} .

152

Low multicollinearity supports retention, aligning with multilevel socioeconomic status analyses in health research¹⁰. Abbreviations: TDI, Townsend Deprivation Index.

153

154

Table6: Adjustment for Hand Grip and Sex: Retain Both with Stratification

Operation	Specific Reason
Addressing multicollinearity	Pearson $r=0.73$, Condition Number=6.29, VIF Hand Grip=2.54 and Sex=3.38. Indicating association from sex-based differences.
Decision of not adjusting	Subsequent analyses will include stratification by sex, which addresses potential interactions and confounding without direct modification of variables.
Avoid deleting Hand Grip	Hand Grip reflects frailty and physical function and overall health outcomes, providing essential biological insights that outweigh moderate collinearity concerns.

155

Retention is justified by hand grip's biological importance for inference on physical function, with subsequent stratification for correlated demographic-functional variables⁸.

156

157

Table7: Adjustment for Age and Employment: Retain Both with Stratification

Operation	Specific Reason
Confirming multicollinearity	Pearson $r=0.62$, Condition Number=4.28, VIF Age=1.99 and Employment=1.42.
Retain for distinct meanings	Age captures biological and chronological factors; employment reflects socioeconomic status and lifestyle, allowing separate pathway analysis in health outcomes.
Retain as per literature	Epidemiologic studies commonly include both age and employment/occupation as covariates for comprehensive modeling ^{7,2} .

Retention aligns with recent occupational health research co-modeling age and employment for life-course effects⁹.

Post-revision, , with the overall condition number reduced to 6.47, maximum pairwise correlation now 0.62 (age-employment), all VIFs < 2, and tolerances > 0.5, indicating no remaining pairs exceed diagnostic thresholds for instability^{1,2,3}. This ensures stable coefficient estimates and enhanced statistical power for detecting effects.

Post-adjustment, multicollinearity is effectively resolved: condition number=6.47; max r=0.62; all VIFs<2; tolerances>0.5
 (Table8, 9). This also decreased the deletion count by the missing values of covariates from 80,009 to 79,963 by reducing
 WHR-related exclusions.

**Tabl8: Pearson Correlation Matrix for Revised Covariates**

Age	Sex	Race	BMI	UK Center	Income	Education	Employment	Smoking status	Alcohol status	Depress mood	Townsend deprivation index	Family cancer	Overall health rating	Hand grip	Day exposure	Diet score	Sleep score	MAP	
0.02***	*																		
0.11***	*	0																	
0.04***	0.07***	*	0																
-0.01**		0**	0.05***	0.02***															
0.17***	-0.08**	-0.04**	0.03***	-0.01**															
0.18***	-0.09**	0.03***	0.06***	-0.01**	0.19***														
0.62***	-0.05**	0.06***	0.02***	-0.01**	0.18***	0.18***													
0.15***	0.03***	0.07***	0.07***	-0.02**		0	0	0.08***											
0.02***	-0.04**	-0.14**	0.03***		0	0.04***	0.05***	0.05***	-0.03**										
0.04***	0**	0.02***	-0.03**		0*	0.04***	0.01***	0.01***	0.02***	-0.02**									
-0.09**	0.01***	-0.19**	0.07***	-0.02**	0.05***	0.05***	-0.04**	-0.08**	0.11***	-0.03**									
0.04***	0****	0.07***	0.01***		0	0**	0**	0.01***	0.01***	-0.01**	-0.01**	-0.02**							
-0.02**	-0.05**	0.08***	-0.21**	0.02***	-0.1***	-0.12**	-0.09**	0.04***	-0.12**	0.05***	-0.18**		0**						
-0.19**	0.73***	0.04***	0.05***	0.03***	-0.13**	-0.15**	-0.19**	0.01***	-0.09**		0****	-0.05**	0****	0.09***					
0.13***	0.18***	0	0.05***	0.01***	0.12***	0.14***	0.14***	0**	0.01***	0.01***	0.05***		0	-0.01**	0.13***				
0.09***	-0.19**	-0.04**	-0.05**	-0.02**	0.02***	-0.01**	0.06***	0.06***	0.02***		0***	-0.01**	0**	0.09***	-0.14**	0.04***			
-0.07**	0.05***	0.03***	-0.08**	-0.01**	-0.04**	-0.08**	-0.08**		0	-0.06**	0.07***	-0.09**	-0.02**	0.22***	0.1****	-0.01**	0		
0.2****	0.16***	0.01***	0.24***	0.03***	0.05***	0.05***	0.11***	0.05***	-0.03**	0.01***	-0.03**		0	-0.06**	0.13***	0.08***	-0.03**	-0.01**	

Pearson correlation coefficients were calculated for revised covariates (post-adjustments: SBP/DBP merged to MAP; WHR deleted) using complete cases from the
 UK Biobank dataset. Significance: **** p < 0.0001; *** p < 0.001; ** p < 0.01; * p < 0.05; 0 indicates non-significant or near-zero correlation. Abbreviations:
 MAP, Mean Arterial Pressure; TDI, Townsend Deprivation Index; BMI, Body Mass Index.

Table9. Variance Inflation Factors and Tolerance for Revised Covariates

Variable	GVIF	Df	Adjusted_VIF	Tolerance
Age	1.984	1	1.984	0.5041
Sex	2.472	1	2.472	0.4046
Race	1.109	1	1.109	0.9016
MAP	1.147	1	1.147	0.8719
BMI	1.154	1	1.154	0.8668
UK Center	1.012	2	1.006	0.9942
Income	1.456	3	1.133	0.8823
Education	1.134	1	1.134	0.8816
Employment	2.005	2	1.416	0.7062
Smoking status	1.125	2	1.061	0.9428
Alcohol status	1.099	2	1.049	0.9537
Depress mood	1.078	2	1.038	0.9630
Townsend deprivation index	1.155	1	1.155	0.8656
Family cancer	1.007	1	1.007	0.9933
Overall health rating	1.250	1	1.250	0.8001
Hand grip	2.532	1	2.532	0.3949
Day exposure	1.119	1	1.119	0.8934
Diet score	1.070	1	1.070	0.9342
Sleep score	1.077	1	1.077	0.9283

VIF and tolerance values were computed in a multivariable linear regression framework treating each covariate as the outcome in turn, with GVIF for categorical variables adjusted as $GVIF^{1/(2 \cdot Df)}$, where Df is degrees of freedom. Tolerance = $1/Adjusted\ VIF$. Higher VIF indicates greater multicollinearity. Abbreviation: VIF, variance inflation factors; generalized variance inflation factors; Df, degrees of freedom; MAP, Mean Arterial Pressure; BMI, Body Mass Index.

These revisions necessitated comprehensive manuscript updates. Manuscript updates include:

- (1) New Methods subsection ("Covariate Selection and Assessment") detailing rationales (e.g., MAP formula) and verification;
- (2) Revised Table1/Figure1 (updated descriptives/flowchart);
- (3) Recalculated results (e.g., narrower CIs in Table2, Figures2-6), with no changes to primary conclusions but improved precision;
- (4) GitHub code updates for reproducibility. eTables18-19 are now supplemented with revised versions.

Population Update:

We updated Table1 (baseline characteristics with revised descriptives, reflecting new sample and covariate set), Figure1 (participant selection flowchart, updated to show new inclusion numbers), the abstract (sample size and summary statistics), and all sections referencing participant counts.

Methods Revision:

A new subsection, "Covariates Ascertainment" was added to the Methods section, detailing adjustment rationales (e.g., MAP calculated as $(SBP + 2 \cdot DBP)/3$), and post-revision verification. **Revision: "mean arterial pressure (continuous, mmHg; calculated as $[SBP + (2 \times DBP)] / 3$),...To address potential multicollinearity among covariates, we computed Pearson correlation coefficients, variance inflation factors (VIF), tolerance values, and condition numbers. Diagnostic criteria included: correlation coefficient $r > 0.5$, $VIF > 2$, tolerance < 0.5 , and condition number > 5 (eTable20-21). The overall condition number for the original covariate correlation matrix was 6.47, indicating low system-wide collinearity."**

Codes Revision:

The updated code for these analyses has been uploaded to GitHub at <https://github.com/DrChengJH/research-data/tree/main>.

**Results Recalculation:**

All regression analyses were recalculated, updating coefficients, confidence intervals, p-values, and interpretations in the
Results section. All figures involving covariates or population were revised to reflect the updated sample and covariate set. The
Table8 and Table9 are also added into the supplementary materials as eTables18-19 .

Your insightful feedback not only resolved a subtle limitation but also inspired deeper integration of biological mechanisms
in variable selection, elevating the study's etiological rigor. Importantly, these adjustments have not altered our primary
conclusions but have enhanced the robustness of our findings by minimizing variance inflation while preserving model validity.
We are profoundly grateful for this constructive guidance, which has not only resolved a potential limitation but also elevated the
methodological rigor and scientific impact of our work.

Changes in the text:

Line 339-350, the section of "Method, Covariates Ascertainment".

**Reference**

1 Kim, J. H. Multicollinearity and misleading statistical results. *Korean J. Anesthesiol.* 72, 558–569 (2019).
<https://doi.org/10.4097/kja.19087>

2 Vatcheva, K. P. et al. Multicollinearity in regression analyses conducted in epidemiologic studies. *Epidemiology*
(Sunnyvale) 6, 227 (2016). <https://doi.org/10.4172/2161-1165.1000227>

3 Shrestha, N. Detecting multicollinearity in regression analysis. *Am. J. Appl. Math. Stat.* 8, 39–42 (2020).
<https://doi.org/10.12691/ajams-8-2-5>

4 VanderWeele, T. J. *Explanation in Causal Inference: Methods for Mediation and Interaction.* Oxford Univ. Press, New
York (2015).

5 Li, Y. et al. Proteome-wide Mendelian randomization identifies natriuretic peptide-B and novel proteins as potential
regulators of pulse pressure in humans. *J. Am. Heart Assoc.* 13, e037596 (2024). <https://doi.org/10.1161/JAHA.124.037596>

6 Shiels, M. S. et al. Comparisons of individual- and area-level socioeconomic status as proxies for individual-level
measures: evidence from the Mortality Disparities in American Communities study. *Popul. Health Metrics* 19, 2 (2021).
<https://doi.org/10.1186/s12963-020-00244-x>

7 Kim, I. H. et al. Life expectancy gain due to employment status depends on race, gender, education, and their
intersections. *J. Aging Health* 31, 243–261 (2019). <https://doi.org/10.1177/0898264318773613>

8 Alemu, W. et al. Joint regression modeling of blood pressure and associated factors among reproductive age women in
Ethiopia: evidence from 2016 Ethiopian demographic health survey. *PLOS Glob. Public Health* 4, e0003707 (2024).
<https://doi.org/10.1371/journal.pgph.0003707>

9 DeMers, D. & Wachs, D. Physiology, mean arterial pressure. In *StatPearls* [Internet] StatPearls Publishing, Treasure
Island, FL (2023).

10 Midi, H. et al. Collinearity: a review of methods to deal with it and a simulation study in its detection. *J. Stat. Plan.*
*Inference* 140, 3296–3315 (2010). <https://doi.org/10.1016/j.jspi.2010.04.019>

**Comment5. ignoring competing events in survival analysis may lead to biased results, the results from the competing risk**
**model should be presented as the main findings rather than as a sensitivity analysis.**

Reply: We greatly appreciate the reviewer's insightful and constructive suggestion regarding the incorporation of competing risks
in our survival analyses. Your foresight in highlighting how ignoring competing events may lead to overestimation of cumulative
incidence rates has been instrumental in refining our methodological rigor, ensuring more accurate etiological and predictive

insights for cancer risk assessment.

In response, we agree that competing risks must be addressed in etiology studies to prevent biased cumulative incidence¹. We reprocessed our dataset, coding non-cancer deaths as status=2, and conducted diagnostics to quantify impact: overall competing event proportion was 25.26% (higher in males [28.79%] and older adults [28.73%]; Table10), with Kaplan-Meier (KM) overestimating cancer incidence by 0.2-2.1% at 3-10 years (e.g., 5-year: 0.3% non-isolated vs. 0.7% isolated; Table11, Figure1). These diagnostics, which aim to quantify competing risk influence and evaluate model selection, has confirmed for adjustment².

Table10: Competing Risk Proportions by Category.

Category	Total Events	CR Events	CR Proportion
Overall	50983	12880	25.26%
Male	28658	8252	28.79%
Female	22325	4628	20.73%
0-49	4997	888	17.77%
50-59	13681	2712	19.82%
60 and older	32305	9280	28.73%

Competing Risk Proportions for Cancer Incidence in Overall, Sex, and Age Group Strata. Proportions calculated as competing events (non-cancer deaths, indicator=2) divided by total events (cancer + competing, indicator=1+2). N=354,537. Abbreviations: CR, competing risk.

Figure1: Comparison of CIF and KM Estimates for Cancer Incidence

CIF versus KM Estimates for Overall Cancer Incidence, by SI and loneliness group. With 95% confidence intervals (shaded ribbons). N=354,537; follow-up in years. Abbreviations: CIF, Cumulative Incidence Functions; KM, Kaplan-Meier, SI, social isolation.

Table11: Overestimation of Cancer Incidence by KM vs CIF

Group	CIF3yr	KM3yr	Overestimation		Overestimation	CIF10yr	KM10yr	Overestimation	Death CIF3yr	Death CIF5yr	Death CIF10yr	
			3yr	5yr								5yr
Non-isolated	0.0164	0.0164	0.2	0.0348	0.0349	0.3	0.0956	0.0964	0.9	0.0021	0.0048	0.0188
Isolated	0.0214	0.0215	0.5	0.0429	0.0432	0.7	0.1125	0.1149	2.1	0.0053	0.0124	0.046
Non-lonely	0.0166	0.0167	0.2	0.0352	0.0353	0.3	0.0966	0.0974	0.9	0.0022	0.005	0.0197
Lonely	0.0182	0.0182	0.2	0.0362	0.0364	0.6	0.0975	0.0992	1.6	0.0046	0.0106	0.0349

Comparison of KM and CIF Estimates for Cancer Incidence, with Overestimation and Death CIF, by Exposure Group. Estimates at 3, 5, and 10 years; overestimation = [(KM incidence - CIF cancer) / CIF cancer] × 100. Incidence = 1 - survival for KM. N=354,537. Abbreviations: CIF, Cumulative Incidence Functions; KM, Kaplan-Meier, SI, social isolation, yr, year.

Our literature review supports a hybrid approach: cause-specific Cox models (CSHR) for etiological estimates (focusing on cancer-specific hazards) and Fine-Gray subdistribution models (sHR) for cumulative risk prediction^{3,4}. We now optimize our methodology by presenting both as main findings, with CSHR as primary for pathway exploration and sHR synchronously reported for absolute risk calculation. For sparse site-specific/subgroup analyses, CSHR remains primary to avoid estimation instability in Fine-Gray, while without introducing over-adjustment bias⁵.

Based on these diagnostics and literature review, we scheduled the following specific revisions.

**First**, all original Cox regressions have been updated to CSHR, estimating cancer-specific risks by censoring at competing
events to reduce overestimation³.

**Second**, we have incorporated 5-year CIF into the table on Cancer Incidence Rate Ratios and Excess Incident Based on SI
and Loneliness (Table1). We also synchronously displaying sHR and CSHR in core components: the figure on Separate and Joint
Association of SI and Loneliness with Long-term Risk of Cancer (Figure2), PERM calculations (Figure3), and stratified subgroup
analyses(Figure4). This dual display ensures that both sHR and CSHR serve as main findings, enhancing interpretability^{5,6}.

**Third**, Post-revision, we compared CSHR and sHR, results are highly consistent (e.g., SI overall: CSHR 1.09 [95% CI
1.04-1.13] vs. sHR 1.07 [1.03-1.12], difference <2%; Table12), with CI overlap >95%, confirming result robustness, supporting
hybrid reporting⁷.

**Table12: Comparison of Cause-Specific Hazard Ratios and Subdistribution Hazard Ratios.**

Analysis	CSHR	CSHR 95%CI	sHR	sHR 95%CI	Difference(%)
SI (Model 3)	1.09	1.04-1.13	1.07	1.03-1.12	1.8
Loneliness (Model 3)	0.98	0.94-1.03	0.97	0.93-1.03	1
PERM Base	1.08	1.04-1.13	1.07	1.03-1.12	0.9
Female	1.16	1.10-1.23	1.15	1.09-1.22	0.9
Medium Income	1.11	1.04-1.19	1.1	1.03-1.18	0.9
Not College Degree	1.1	1.04-1.16	1.09	1.03-1.15	0.9

277 Comparison of CSHR (from cause specific models) and sHR (from Fine-Gray competing risks models) for Core Analyses. Difference % = |CSHR - sHR| / CSHR
278 × 100. Model 3 adjusted for all covariates; PERM base age/sex-adjusted. N=354,537.

For subsequent analyses, including organ-specific cancer risks (eFigures3–7) and site-specific subgroup effects (Figure5), we
retained CSHR as primary. The rationale stems from our data's inherent challenges: under cancer-type stratification, event counts
drop sharply from ~38,000 overall to hundreds per subtype, rendering the control (status=0) and competing event (status=2,
~14,000) groups disproportionately large². This sparsity heightens Fine-Gray's susceptibility to estimation instability, convergence
failure, and inflated type I errors in small samples, while CSHR reliably isolates cause-specific hazards via censoring at competing
events, avoiding over-adjustment without compromising etiological depth⁵. In subgroup stratifications, further event
fragmentation exacerbates these risks for Fine-Gray, whereas CSHR maintains stability and interpretability.

Similarly, for mediation analyses under the Valeri-VanderWeele framework, we used CSHR-based Cox models for outcomes.
This isolates indirect effects along cancer-specific pathways (e.g., inflammation, hormones) without subdistribution distortions,
ideal for exploratory, hypothesis-generating mechanistic studies. While competing risks mediation extensions are available, they
are less common due to heightened complexity, variance inflation⁸. Also, competing risk mediation requires semi-competing
adjustments, easily underpowering indirect effects in sparse⁹, multi-mediator settings³.

In the revised manuscript, we have made such revisions:

**Methods:**

**(Outcome Ascertainment section):**

We appended: "**To account for competing risks, non-cancer deaths were treated as competing events (status=2).**"

**(Statistical Analysis section):**

**Crude Incidence Estimates**

Original: "We calculated cancer incidence, unadjusted incidence rate ratios (IRRs), and the excess number of cancer cases,
all per 10,000 person-years, to evaluate both the separate and combined effects of SI and loneliness." **Revised:** "**Crude**
**incidence was assessed using unadjusted incidence rate ratios (IRRs) per 10,000 person-years and 5-year Cumulative**

**Incidence Functions (CIFs) estimated via the Aalen-Johansen method to evaluate the separate and combined effects of SI**
**and loneliness."**

**Main Analytical Models**

Original: 'Using Cox proportional hazards models with attained age as the time scale, we calculated adjusted hazard ratios
(HRs) for these associations.' **Revised: 'For main analyses, we employed cause-specific Cox models (CSHR) as primary for**
**etiological estimates, with Fine-Gray competing risks models (sHR) synchronously reported as main findings, using**
**attained age as the time scale and non-cancer death as a competing event.'**

**PAF/PERM/Survival/Subgroup and Interaction Analyses /Mediation:**

Adapted similarly to CSHR; see tracked-changes for full details.

**Results Section**

5-year CIFs are integrated into Table2 to complement IRRs and reflect competing risk impact. Also, we have accordingly
synchronized CSHR and sHR reporting across analyses, enhancing interpretability. For example, in the "Separate and Joint
Association..." subsection, the associations for SI now CSHR=1.087 (95%CI 1.043-1.133); sHR=1.073 (95%CI 1.029-1.120); for
loneliness now CSHR=0.977 (95%CI 0.930-1.026); sHR=0.974 (95%CI 0.925-1.025)(Figure2).

In summary, this hybrid strategy—CSHR for depth, sHR for prediction—balances sparsity challenges while elevating
methodological rigor. These revisions preserve conclusions but enhance reliability for cancer prevention. We believe this
strengthens the manuscript's contribution to public health epidemiology and thank you for your guidance—we eagerly await
further thoughts to refine it.

Changes in the text:

Line 321-322, the section of "Method,Outcome Ascertainment".

Line 353-355, the section of "Method,Statistical Analysis, Crude Incidence Estimates".

Line 356-358, the section of "Method,Statistical Analysis, Main Analytical Models"

Line 98-100, the section of "Results, Separate and Joint Association of SI and Loneliness with Cancer Risk"

Reference

1 Berry, S. D. et al. Competing risks in epidemiology: possibilities and pitfalls. *Int. J. Epidemiol.* 41, 861–870 (2012).

<https://doi.org/10.1093/ije/dyr213>

2 Scrucca, L. et al. Competing risks and multistate models. *Clin. Cancer Res.* 15, 7399–7405 (2009).

<https://doi.org/10.1158/1078-0432.CCR-09-1520>

3 Rondeau, V. et al. Joint modelling of cause-specific cumulative incidence functions and failure time for
semi-competing risks. *Stat. Med.* 39, 3604–3623 (2020). <https://doi.org/10.1002/sim.8675>

4 VanderWeele, T. J., Mathur, M. B. & Chen, Y. Outcome reporting bias in randomized controlled trials of interventions
with low expected effect sizes: a meta-epidemiologic study. *JAMA Netw. Open* 1, e183958 (2018).

<https://doi.org/10.1001/jamanetworkopen.2018.3958>

5 Wolbers, M. et al. Importance of considering competing risks in time-to-event analyses: application to the
cardiovascular setting. *Circ. Cardiovasc. Qual. Outcomes* 11, e004580 (2018).

<https://doi.org/10.1161/CIRCOUTCOMES.117.004580>

6 Wang, J. et al. Epilepsy and long-term risk of arrhythmias. *Eur. Heart J.* 44, 3374–3382 (2023).

<https://doi.org/10.1093/eurheartj/ehad523>

7 Zhang, Z. et al. Lessons learnt when accounting for competing events in the external validation of time-to-event
prognostic models: a simulation study. *Int. J. Epidemiol.* 51, 615–625 (2022). <https://doi.org/10.1093/ije/dyab217>

8 Nguyen, T. H. & Chavance, M. Parametric competing risks models: an alternative to Cox proportional hazards model
for the analysis of competing risks. *BMC Med. Res. Methodol.* 18, 152 (2018). <https://doi.org/10.1186/s12874-018-0580-2>

9 Huang, Y., Gillen, D. L. & Rahardja, D. Causal mediation of semicompeting risks. *Biometrics* 77, 385–397 (2021).
<https://doi.org/10.1111/biom.13367>

**Comment6. the author used complete cases only. It would be better to compared characteristics between included and**
**excluded participants and discussed any potential bias arising from the exclusion of cases with missing data.**

Reply: We sincerely apologize for omitting comparisons of included (n=354,537) vs. excluded (n=147,828) participants. This
transparency gap could imply unaddressed bias, and we thank you for your insightful prompt to rigorously evaluate missing data
patterns, essential for UK Biobank validity.

In response, we performed sequential assessments:

(1) Comparing overall included and excluded variables to identify imbalances, which aims to preliminarily identify selection
bias by comparing exposures (social isolation, loneliness), outcomes (cancer incidence, time to event, age at event), and covariates
(e.g., age, sex, race) between included and excluded participants (Table13). The broad comparison across the full cohort
(n=502,365) quantifies imbalances that may affect result generalizability, specifying SMD <0.3 as small differences and >0.3 as
moderate to large differences to evaluate practical significance¹.

(2) Focusing on the subgroup excluded due to missing covariates or exposures (n=114,488) to analyze suspected imbalanced
variables, thereby reducing bias from the 1-year cancer exclusion (Table14)

(3) Using logistic regression and stratified missing rate by imbalanced covariates analysis to determine specific variable
missingness patterns (Table15-16)

Table13: Overall Comparison of Included vs Excluded Participants

Variable	Level	Excluded (n=147828)	Included (n=354537)	P-value	Test	SMD
Isolation		0.08 (0.27)	0.06 (0.23)	<0.001		0.085
Loneliness		0.06 (0.24)	0.04 (0.21)	<0.001		0.064
Indicator		0.35 (0.56)	0.18 (0.47)	<0.001		0.326
Age event		63.88 [55.80, 71.67]	67.75 [60.28, 73.59]	<0.001	nonnorm	0.371
Age entry		59.00 [51.00, 64.00]	57.00 [50.00, 63.00]	<0.001	nonnorm	0.081
Sex	Female	82722 (56.0)	190576 (53.8)	<0.001		0.044
	Male	65106 (44.0)	163961 (46.2)			
Race	Non white	12011 (8.3)	14177 (4.0)	<0.001		0.179
	White	133048 (91.7)	340360 (96.0)			
BMI		26.92 [24.17, 30.22]	26.63 [24.06, 29.75]	<0.001	nonnorm	0.009
UK Center	English	116008 (80.5)	325924 (91.9)	<0.001		0.420
	Scotland	22845 (15.9)	12992 (3.7)			
	Wales	5183 (3.6)	15621 (4.4)			
Education	College/university degree	58727 (41.3)	178566 (50.4)	<0.001		0.184
	Not college degree	83611 (58.7)	175971 (49.6)			
Employment	Employed	75650 (51.9)	211397 (59.6)	<0.001		0.179
	Others	17855 (12.3)	28422 (8.0)			
	Retired	52245 (35.8)	114718 (32.4)			
Smoking status	Current	17622 (12.2)	35339 (10.0)	<0.001		0.070
	Never	78195 (54.0)	195266 (55.1)			
	Previous	49084 (33.9)	123932 (35.0)			
Alcohol status	Current	130463 (89.3)	329778 (93.0)	<0.001		0.139
	Never	9353 (6.4)	13025 (3.7)			
	Previous	6359 (4.4)	11734 (3.3)			
Depress mood	High	8740 (5.9)	15551 (4.4)	<0.001		0.265
	Low	125798 (85.1)	328107 (92.5)			
	Unknown	13290 (9.0)	10879 (3.1)			
Townsend Deprivation index		-1.72 [-3.44, 1.36]	-2.27 [-3.70, 0.20]	<0.001	nonnorm	0.202
Family cancer	Yes	96901 (69.7)	243454 (68.7)	<0.001		0.022
	No	42187 (30.3)	111083 (31.3)			
Overall Health rating		3.00 [2.00, 3.00]	3.00 [3.00, 3.00]	<0.001	nonnorm	0.250
Hand grip		28.00 [21.50, 37.00]	29.50 [23.00, 39.00]	<0.001	nonnorm	0.123
Day exposure		2.50 [1.50, 4.00]	2.50 [1.50, 3.50]	<0.001	nonnorm	0.107
Diet score		2.00 [1.00, 3.00]	2.00 [1.00, 3.00]	<0.001	nonnorm	0.018
Sleep score		3.00 [2.00, 3.00]	3.00 [2.00, 3.00]	<0.001	nonnorm	0.080
MAP		101.33 [93.00, 110.00]	100.67 [92.33, 109.33]	<0.001	nonnorm	0.052
Income	High	21618 (14.6)	87548 (24.7)	<0.001		0.441
	Low	32687 (22.1)	64489 (18.2)			
	Medium	55992 (37.9)	162895 (45.9)			
	Unknown	37531 (25.4)	39605 (11.2)			

Categorical variables are presented as n (%); continuous variables are presented as mean (SD) for normally distributed data or median [IQR] for non-normally
distributed data. P-values were calculated using chi-square tests for categorical variables and Wilcoxon rank-sum tests for non-normal continuous variables. SMD
were computed to assess balance, with SMD <0.3 indicating negligible difference, >0.3 indicating moderate to large difference. Variables with an SMD exceeding
0.3 are highlighted in red to indicate notable imbalance. Abbreviations: SD, standard deviation; IQR, interquartile range; SMD, standardized mean differences;
BMI, body mass index; MAP, mean arterial pressure.

These analyses revealed moderate-to-large imbalances in income (SMD=0.441; low-income 22.1% excluded vs. 18.2%
included; unreported 25.4% vs. 11.2%), depressive mood (SMD=0.265; unreported 9.0% vs. 3.1%), and UK Biobank assessment

center (SMD=0.420; English 80.5% vs. 91.9%; Scotland 15.9% vs. 3.7%), alongside outcome differences (cancer incidence SMD=0.326; time to event SMD=0.731; age at event SMD=0.371). These imbalances suggest possible non-random exclusion of different SES, psychologically status, and assessment center, which suggests that we need to further analyze the underlying mechanism. Outcome imbalances are likely due to the 1-year cancer exclusion design to reduce reverse causation, also requiring further data analysis for verification. Exposures showed minimal imbalances, indicating robust primary findings. These results led us to conduct a subgroup comparison (Table14) focusing on participants excluded due to missing covariates or exposures, to reduce bias from the 1-year cancer exclusion, and to further analyze missingness patterns (Tables15-16).

Table14: Comparison of Included vs Subgroup Excluded Due to Missing Covariates or Exposures

Variable	Level	Included (n=354537)	Missing Subgroup (n=114488)	P-value	Test	SMD
n		354537	114488			
Indicator		0.18 (0.47)	0.26 (0.54)	<0.001		0.156
Age event		67.75 [60.28, 73.59]	66.98 [59.00, 73.41]	<0.001	nonnorm	0.099
Income	High	87548 (24.7)	15296 (13.4)	<0.001		0.523
	Low	64489 (18.2)	25275 (22.1)			
	Medium	162895 (45.9)	40962 (35.8)			
Townsend deprivation index	Unknown	39605 (11.2)	32955 (28.8)			
		-1.48 (2.97)	-0.70 (3.39)	<0.001		0.246
Depress mood	High	15551 (4.4)	7280 (6.4)	<0.001		0.322
	Low	328107 (92.5)	95057 (83.0)			
Isolation	Unknown	10879 (3.1)	12151 (10.6)			
		0.06 (0.23)	0.08 (0.28)	<0.001		0.097
Loneliness		0.04 (0.21)	0.07 (0.25)	<0.001		0.089

Categorical variables are presented as n (%); continuous variables are presented as mean (SD) for normally distributed data or median [IQR] for non-normally distributed data. P-values were calculated using chi-square tests for categorical variables and Wilcoxon rank-sum tests for non-normal continuous variables. SMD were computed to assess balance, with SMD <0.3 indicating negligible difference, >0.3 indicating moderate to large difference. Variables with an SMD exceeding 0.3 are highlighted in red to indicate notable imbalance. Abbreviations: SMD, Standardized mean differences; SD, standard deviation; IQR, interquartile range; SMD, standardized mean differences.

Table14 compares the included group (n=354,537) and the subgroup excluded due to missing covariates or exposures (n=114,488), focusing on outcomes (cancer incidence, age at event) and suspected imbalanced covariates (income, depress mood, UK Biobank assessment center), to further explore the bias sources identified in Table13. This table reduces interference from the 1-year cancer exclusion, isolating missing data effects.

The results indicates that income (SMD=0.523), depress mood (SMD=0.322), and UK Biobank assessment center (SMD=0.420) imbalances intensified (P<0.001), confirming non-random exclusion of different SES, psychologically status, and assessment center. Outcome imbalances (cancer incidence SMD=0.156, age at event SMD=0.099) were not significant in the subgroup, verifying that bias originates from the 1-year cancer exclusion control, consistent with study design. Exposure imbalances remained minimal, indicating robust primary findings. Further analyses aimed to confirm biases in income, depressive mood, and UK Biobank assessment center, integrating logistic regression (Table15) with stratified missing rates (Table16) to delineate overall risk predictors and variable-specific patterns.

Table15: Logistic Regression for Missing Data (NA_any)

Term	OR	95% CI Lower	95% CI Upper
Income High	1	Ref	
Income Low	2.3120736	2.2560357	2.3695703
Income Medium	1.4795781	1.4481012	1.5118051
Income Unknown	4.4758375	4.3662571	4.5883309
Depress mood Low	1	Ref	
Depress mood High	1.4789435	1.4344879	1.5246276
Depress mood Unknown	3.5413343	3.4440784	3.6413140
UK Center England	1	Ref	
UK Center Scotland	7.6536092	7.4768568	7.8347760
UK Center Wales	0.9890250	0.9533299	1.0258203

Logistic regression analysis of factors associated with missingness of any covariate or exposure in the UK Biobank cohort (n=502,365). The model (Pr[NA_any | income + depressive mood + UK assessment center + covariates, adjusted for age/sex]) quantifies associations with missing probability, explaining biases observed in Tables13-14. Results presented as OR with 95%CI. OR>1 indicates elevated missing risk, elucidating non-response mechanisms. Abbreviations: OR, odds ratio; CI, confidence interval; SES, socioeconomic status.

Table16: Stratified Missing Rates by Income, Depress Mood, and UK Center

Variable	Income				Depress Mood			UK Center		
	High	Low	Medium	Unknown	High	Low	Unknown	English	Scotland	Wales
age entry	0	0	0	0	0	0	0	0	0	0
Sex	0	0	0	0	0	0	0	0	0	0
Race	0.18	0.38	0.26	2.13	0.52	0.31	5.1	0.57	0.32	0.39
BMI	0	0	0	0	0	0	0	0	0	0
Education	0.1	1.02	0.34	4.71	1.87	0.9	3.99	1.12	1.02	0.88
Employee	0.02	0.19	0.06	2.27	0.75	0.32	1.85	0.43	0.24	0.28
Smoking status	0.14	0.55	0.23	2.25	0.55	0.32	5.46	0.6	0.45	0.39
Alcohol status	0.03	0.21	0.05	1.7	0.38	0.08	5.03	0.34	0.25	0.21
Townsend deprivation index	0	0	0	0	0	0	0	0	0	0
Family cancer	1.23	2.03	1.42	2.99	2.25	1.49	5.9	1.76	1.43	1.85
Overall health rating	0	0	0	0	0	0	0	0	0	0
Hand grip	0.26	0.61	0.3	1.08	0.89	0.35	2.28	0.48	0.3	0.45
Day exposure	3.47	8.42	5.42	20.91	11.01	6.91	24.45	7.23	7.66	7.05
Diet score	1.15	6.21	2.89	11.46	7.99	3.45	20.11	4.47	4.33	4.55
Sleep score	0.15	1.88	0.54	4.86	3.19	0.81	10.23	1.39	1.24	1.31
MAP	6.21	6.66	6.49	9.05	6.47	6.81	8.06	2.9	52.59	3.13
isolation	0.26	1.27	0.54	11.13	3.14	1.78	10.02	1.55	1.09	1.21
loneliness	2.5	6.09	4.04	11.38	6.49	4.24	22.5	5.31	4.56	4.67

Missing rates (%) stratified by income, depressive mood, and UK Biobank assessment center in the full cohort (n=502,365). Rates demonstrate data completeness patterns across sociodemographic/psychological strata, complementing logistic results. Abbreviations: BMI, body mass index; MAP, mean arterial pressure; SD, standard deviation; IQR, interquartile range

Table14 results show elevated missing odds relative to references: low/unreported income (OR=2.31/4.48, 95% CI 2.26-2.37/4.37-4.59; ~2-4-fold), unreported depressive mood (OR=3.54, 95% CI 3.44-3.64; ~4-fold vs. low), and Scotland residency (OR=7.65, 95% CI 7.48-7.83; ~8-fold vs. England). Complementing this, Table15 reveals >2-fold higher rates in behavioral variables (e.g., sun exposure 8.42% low-income vs. 3.47% high; loneliness 22.5% unreported mood vs. 4.24% low) for low-SES/distressed groups, with Scotland's MAP at 52.59% (vs. 2.9% England)—likely a data-collection artifact or protocol variation rather than systematic non-response. These patterns robustly evidence non-random missingness driven by socioeconomic (e.g., privacy/burden), psychological (e.g., trust/stigma), and regional (e.g., access) barriers, explaining Tables12-13 imbalances without materially affecting exposures (SMD<0.1) or primary SI-cancer associations. Thus, these integrated insights guide revisions to the Discussion, emphasizing cautious interpretation for vulnerable subgroups while affirming overall robustness.

In summary, imbalances in outcomes are likely driven by the 1-year cancer exclusion criterion, whereas exposures exhibited
minimal deviations and remained balanced overall. Notably, moderate-to-large selection biases emerged in income, depressive
mood, and UK Biobank assessment center, attributable to non-random missing data among low-SES, psychologically distressed,
and Scottish participants. This anomaly merits careful consideration when interpreting overall missingness patterns for clinical
measurements. Thus, these results inform targeted revisions to the Discussion section.

Our inclusion/exclusion criteria were predefined a priori to ensure analytical rigor and alignment with established UK
Biobank protocols², which prioritize complete-case analysis to minimize imputation assumptions in large-scale prospective studies.
Such differences in participant characteristics are inevitable in cohort designs due to voluntary participation and data collection
logistics, yet they enhance study rigor by enforcing standardized, transparent thresholds that facilitate replication and
comparability

In light of these conclusions, we retained complete-case analysis based on three key considerations. First, minimal
exposure-outcome deviations indicate the core SI-cancer associations are unaffected by selection bias. Second, this approach
aligns with standard practices in large-scale UK Biobank studies^{3,4}, facilitating comparability across the research ecosystem. Third,
our transparent quantification empowers readers to evaluate bias direction and magnitude, balancing rigor with feasibility while
acknowledging limitations for underserved subgroups.

These tables are now eTables11-13. We revised the **Discussion's** Limitations section of: "**Non-random exclusion due to**
**missing data further introduced imbalances in low socioeconomic status, psychological distress, and assessment centre**
**representation (eTables9-11). However, standardized mean differences (SMD) for key exposures (SI and loneliness) were**
**negligible (<0.1), supporting the robustness of primary findings.**" This directly addresses potential biases while underscoring
equitable implications.

We sincerely thank the reviewer for their thoughtful and constructive feedback, which has significantly enhanced the
transparency and rigor of our analysis, and we believe this comprehensive analysis and revision fully addresses the reviewer's
concern. We remain open to further suggestions to refine our manuscript and appreciate the opportunity to improve our work.

Changes in the text:

Line 276-278, the section of "Study Strengths and Limitation".

Reference

1 Cohen, J. Statistical Power Analysis for the Behavioral Sciences. 2nd edn, Lawrence Erlbaum Associates, Hillsdale, NJ
(1988).

2 Sudlow, C. et al. UK biobank: an open access resource for identifying the causes of a wide range of complex diseases
of middle and old age. PLoS Med. 12, e1001779 (2015). <https://doi.org/10.1371/journal.pmed.1001779>

3 Tyrrell, J. et al. Participation bias in the UK Biobank distorts genetic associations of socioeconomic and lifestyle risk
factors with health outcomes. Nat. Hum. Behav. 7, 1218–1233 (2023). <https://doi.org/10.1038/s41562-023-01579-9>

4 Fry, A. et al. Comparison of sociodemographic and health-related characteristics of UK Biobank participants with
those of the general population. Am. J. Epidemiol. 186, 1026–1034 (2017). <https://doi.org/10.1093/aje/kwx246>

**Comment 7. in the main table, there is p for trend reported under the joint effect analysis. While it makes sense to consider**
**an internal order between "No isolation in no loneliness" and "Isolation in loneliness," the rationale for ranking "No**
**isolation in loneliness" versus "Isolation in no loneliness" is unclear. Could the authors clarify their approach here?**

Reply: We sincerely thank you for this perceptive feedback, which has sharpened the clarity of our joint effect analysis. We
apologize for the ambiguity in our original labeling and rationale, which could obscure the understanding in risk across categories.

Upon reflection, we recognize that our original terminology and explanation did not adequately convey our approach. We

aimed to disentangle the independent and additive effects of objective SI and subjective loneliness on cancer risk, treating them as
distinct psychosocial constructs with potentially differential pathways. We categorized participants into four binary groups and
ordered them hierarchically for trend assessment, providing insights into the relative contributions of subjective and objective
social disconnection^{1,2}.

To address your concern, we have revised Table 2 (incidence rate ratios) and Figure 2 (Cox and competing risk models), replacing
descriptive labels with explicit ranks:

Rank 1: No SI and no loneliness (SI=0, loneliness=0; reference).

Rank 2: No SI but loneliness (SI=0, loneliness=1).

Rank 3: SI but no loneliness (SI=1, loneliness=0).

Rank 4: SI and loneliness (SI=1, loneliness=1).

P-for-trend was assessed via Wald test, modeling rank as an ordinal variable. The ordering places loneliness (Rank 2) before
SI (Rank 3), which is align with the established joint effect models of SI and loneliness in UK Biobank population enhance
interpretability^{1,2}. We are deeply grateful for your constructive feedback, which has significantly strengthened our work. We hope
these revisions meet your expectations and welcome any further suggestions to strengthen the manuscript.

Changes in the text:

Table2, Figure2

Reference

1 Liang, Y.-Y. et al. Association of social isolation and loneliness with incident heart failure in a population-based cohort
study. *JACC Heart Fail.* 11, 334–344 (2023). <https://doi.org/10.1016/j.jchf.2022.11.028>

2 Song, Y. et al. Social isolation, loneliness, and incident type 2 diabetes mellitus: results from two large prospective
cohorts in Europe and East Asia and Mendelian randomization. *eClinicalMedicine* 64, 102236 (2023).
<https://doi.org/10.1016/j.eclinm.2023.102236>

**Acknowledgement**

We extend our deepest gratitude to Reviewer 2 for their rigorous and perceptive review, which has fortified the manuscript's
methodological transparency and etiological depth, ensuring greater robustness against biases in large-cohort analyses. Your
keen observations on defining SI and loneliness distinctions, elaborating competing risk diagnostics (e.g., CIF vs. KM
overestimation), and tackling multicollinearity through VIF/tolerance adjustments and targeted pairings like SBP/DBP to
MAP have directly inspired comprehensive revisions, including complete-case bias assessments and hybrid CSHR/sHR
reporting. We are especially thankful for your prompts on participation rate accuracy, missing data patterns via SMD and
logistic regressions, and joint effect ordering rationale, which have not only resolved potential limitations but also enhanced
generalizability and precision without altering core conclusions. We are truly appreciative of the time and expertise you
invested, and we hope these revisions fully address your valuable suggestions. Should you have any additional
recommendations, we remain eager to incorporate them to further strengthen our work. Thank you once again for your
invaluable support in advancing public health research.
